# Optimizing Canaries for Privacy Auditing with Metagradient Descent

**Matteo Boglioni**[*]
ETH Zurich
mboglioni@ethz.ch

**Terrance Liu**[*]
Carnegie Mellon University
terrancl@cmu.edu

**Andrew Ilyas**
Carnegie Mellon University
Stanford Statistics
andrewi@stanford.edu

**Zhiwei Steven Wu**
Carnegie Mellon University
zstevenwu@cmu.edu

## Abstract

In this work we study *black-box privacy auditing*, where the goal is to lower bound the privacy parameter of a differentially private learning algorithm using only the algorithm's outputs (i.e., final trained model). For DP-SGD (the most successful method for training differentially private deep learning models), the canonical auditing approach uses *membership inference*—an auditor comes with a small set of special "canary" examples, inserts a random subset of them into the training set, and then tries to discern which of their canaries were included in the training set (typically via a membership inference attack). The auditor's success rate then provides a lower bound on the privacy parameters of the learning algorithm. Our main contribution is a method for *optimizing* the auditor's canary set to improve privacy auditing, leveraging recent work on metagradient optimization (Engstrom et al., 2025). Our empirical evaluation demonstrates that in certain instances, using such optimized canaries can improve empirical lower bounds for differentially private image classification models by several times when compared to canaries proposed in prior work. Furthermore, we demonstrate that our method is *DP-SGD agnostic* and *efficient*: canaries optimized for non-private SGD with a small model architecture remain effective when auditing larger models trained with DP-SGD.

## 1 Introduction

Differential privacy (DP) (Dwork et al., 2006) offers a rigorous mathematical framework for safeguarding individual data in machine learning. Within this framework, differentially private stochastic gradient descent (DP-SGD) (Abadi et al., 2016) has emerged as the standard for training differentially private deep learning models. Although DP-SGD provides theoretical upper bounds on privacy loss based on its hyperparameters, these guarantees are likely conservative, which means they tend to overestimate the privacy leakage in practice (Nasr et al., 2023). In many cases, however, they may not reflect the true privacy leakage that occurs during training. To address this gap, researchers have developed empirical techniques known as privacy audits, which aim to establish lower bounds on privacy loss. In addition to quantifying real-world leakage, privacy auditing can also help detect bugs or unintended behaviors in the implementation of private algorithms (Tramer et al., 2022).

Providing a lower bound on the privacy leakage of an algorithm typically requires the auditor to guess some private information (*membership inference*) using a set of examples (also referred to as *canaries*). For example, in one-run auditing procedures (Steinke et al., 2023; Mahloujifar et al., 2024) (which we discuss further in Section 2.1.2), a random subset of these canaries is inserted into the training dataset, and once the model is trained, the auditor guesses which of these samples belong to that subset. While recent work has made significant progress in tightening these bounds through privacy auditing, the strongest results typically assume unrealistic levels of access or control of the

---

[*]Equal contribution

private training process (broadly speaking, such settings fall under the term *white-box* auditing (Nasr et al., 2023)). In contrast, this work focuses on a more practical and restrictive *black-box* setting. Here, the auditor can only insert a subset of carefully crafted examples (called the *canaries*) into the training set and observe the model's output at the *last iterate* (without access to intermediate model states or gradient computations). In other words, the goal of black-box DP auditing reduces to performing membership inference on the canaries based on the *final model output*.

In this work, we study how to optimize canary samples for the purpose of black-box auditing in differentially private stochastic gradient descent (DP-SGD). Leveraging metagradient descent (Engstrom et al., 2025), we introduce an approach for crafting canaries specifically tailored for insertion into the training set during DP auditing. Through empirical evaluation on single-run auditing protocols for DP image classification models, we find that our method consistently yields canaries that surpass standard baselines by more than a factor of two in certain regimes. Notably, our algorithm is computationally efficient: although it involves running (non-private) SGD on a lightweight ResNet-9 architecture, the resulting canaries demonstrate strong performance even when deployed in larger models, such as Wide ResNets, trained under DP-SGD. Furthermore, this improvement persists whether DP-SGD is used for end-to-end training or private finetuning on pretrained networks.

## 1.1 RELATED WORK

Early works in DP auditing (Ding et al., 2018; Bichsel et al., 2018) introduce methods that detect violations of formal DP guarantees, relying on a large number of runs to identify deviations from expected behavior. These techniques, however, are not directly applicable to the domain of differentially private machine learning, as they were developed for auditing simpler DP mechanisms. To tackle this issue, Jagielski et al. (2020) and Nasr et al. (2021) introduce new approaches based on membership inference attacks (MIA) to empirically determine privacy lower bounds for more complex algorithms like DP-SGD. Membership inference consists of accurately determining whether a specific sample was part of the model's training dataset. If the guesser (i.e. *attacker*) can reliably make accurate guesses, it suggests that the model retains information about individual samples observed during training, thereby compromising individuals' privacy. Hence, MIA can be used as a practical DP auditing tool in which lower bounds on how much privacy leakage has occurred can be directly estimated from the success rate of the attacker.

**One-run auditing.** The first auditing methods for DP-SGD relied on many runs of the algorithm, making auditing very expensive and often impractical. To remedy this issue, Steinke et al. (2023); Mahloujifar et al. (2024) reduce the computational cost of auditing by proposing procedures that require only one training run. Kazmi et al. (2024) and Liu et al. (2025) further study how to incorporate stronger MIA methods to empirically improve auditing in this one-run setting. Similarly, Keinan et al. (2025) study the theoretical maximum efficacy of one-run auditing.

**Last-iterate auditing.** Our work builds on the aforementioned one-run auditing methods and focuses specifically on the *last-iterate* auditing regime, which restricts the auditor's access to just the final model weights after the last iteration of DP-SGD. Related work to this regime from Muthu Selva Annamalai (2024) investigates whether the analysis on the last iteration can be as tight as analysis on the sequence of all iterates. Meanwhile, Nasr et al. (2025) propose a heuristic that predicts empirical lower bounds derived from auditing the last iterate. Other works instead focus on partial relaxations of the problem: Cebere et al. (2025) assume that the auditor can inject crafted-gradients, and Muthu Selva Annamalai & De Cristofaro (2024) audit models that are initialized to worst-case parameters.

**Canaries Optimization.** Rather than proposing new auditing procedures, our work studies how to make existing ones more effective by focusing on optimizing canary sets for privacy auditing. Similarly, Jagielski et al. (2020) develop a method that uses singular value decomposition to obtain canaries more robust to gradient clipping. Nasr et al. (2023) evaluate various procedures to optimize canaries for their *white-box* auditing experiments. To better audit differentially private federated learning, Maddock et al. (2023) craft an adversarial sample that is added to a client's dataset used to send model updates. Finally, in the context of auditing LLMs, Panda et al. (2025) proposes using tokens sequences not present in the training dataset as canaries, while Meeus et al. (2025) create canaries with a low-perplexity, in-distribution prefixes and high-perplexity suffixes.

**Metagradient computation.** Our work also makes use of recent advancements in computing *metagradients*, gradients of machine learning models' outputs with respect to their hyperparameters or other quantities decided on prior to training. Prior work on metagradient computation falls under two categories: *implicit differentiation* (Bengio, 2000; Koh & Liang, 2017; Rajeswaran et al., 2019; Finn et al., 2017; Lorraine et al., 2020; Chen & Hsieh, 2020; Bae et al., 2022) aims to approximate the metagradient. On one hand, approximating metagradients allows for scalability to large-scale metagradient computation; on the other, this approach loosens correctness guarantees and imposes restrictions on what learning algorithms can be used. In contrast, *explicit differentiation* directly computes metagradients using automatic differentiation. However, these works (Maclaurin et al., 2015; Micaelli & Storkey, 2021; Franceschi et al., 2017; Liu et al., 2018) are limited by their scalability to larger models and number of hyperparameters and by numerical instability. We leverage recent work by Engstrom et al. (2025), which takes the explicit approach, but addresses the aforementioned issues by proposing a scalable and memory-efficient method to compute metagradients.

## 2 PRELIMINARIES

Informally, differential privacy provides bounds on the extent to which the output distribution of a randomized algorithm $\mathcal{M}$ can change when a data point is removed or swapped out.

**Definition 2.1** ((Approximate-) Differential Privacy (DP) (Dwork et al., 2006))**.** A randomized algorithm $\mathcal{M} : \mathcal{X}^N \to \mathbb{R}$ satisfies $(\varepsilon, \delta)$-differential privacy if for all neighboring datasets $D, D'$ (i.e., all $D, D'$ such that $|D' \setminus D| = 1$) and for all outcomes $S \subseteq \mathbb{R}$ we have

$$P(\mathcal{M}(D) \in S) \leq e^\varepsilon P(\mathcal{M}(D') \in S) + \delta$$

In the context of machine learning, $\mathcal{M}$ would be a learning algorithm, and this definition requires the model to be insensitive to the exclusion of one training data point. In essence, it bounds the change in the output distribution of the model when trained on neighboring datasets. This implies that the model does not overly depend on any single sample observed.

Since the seminal work of Dwork et al. (2006), various relaxations of differential privacy have been proposed. Below, we define $f$-differential privacy, which we later reference when describing the auditing procedure proposed by Mahloujifar et al. (2024).

**Definition 2.2** ($f$-Differential Privacy (Dong et al., 2022))**.** A mechanism $\mathcal{M}$ is $f$-DP if for all neighboring datasets $\mathcal{S}, \mathcal{S}'$ and all measurable sets $T$ with $|\mathcal{S} \triangle \mathcal{S}'| = 1$, we have

$$\Pr[\mathcal{M}(\mathcal{S}) \in T] \leq \bar{f}\left(\Pr[\mathcal{M}(\mathcal{S}') \in T]\right). \tag{1}$$

Importantly, $f$-DP relates to approximate DP in the following way:

**Proposition 1.** A mechanism is $(\varepsilon, \delta)$-DP if it is $f$-DP with respect to $\bar{f}(x) = e^\varepsilon x + \delta$, where $\bar{f}(x) = 1 - f(x)$.

While a wide range of methods for adding differentially private guarantees to machine learning algorithms have been proposed over the years, DP-SGD (Abadi et al., 2016) has been established as one of the de facto algorithms for training deep neural networks with DP. At a high-level, DP-SGD makes SGD differentially private by modifying it in the following ways: (1) gradients are clipped to some maximum Euclidean norm and (2) random noise is added to the clipped gradients prior to each update step. In Algorithm 5, we present DP-SGD in detail.

### 2.1 AUDITING DIFFERENTIAL PRIVACY

Differentially private algorithms are accompanied by analysis upper bounding the DP parameters $\varepsilon$ and $\delta$. Privacy auditing instead provides an empirical *lower bound* on these parameters. In this work, we focus on a specific formulation of privacy audits: *last-iterate, black-box, one-run* auditing.

#### 2.1.1 LAST-ITERATE BLACK-BOX AUDITING

Our work focuses on *last-iterate black-box* auditing, where the auditor can only insert samples (i.e., canaries) into the training set and can only access the resulting model at the final training iteration.

---

**Algorithm 1:** Black-box Auditing - One Run (Steinke et al., 2023)

---

**Input:** probability threshold $\tau$, privacy parameter $\delta$, training algorithm $\mathcal{A}$, dataset $D$, set of $m$ canaries $C = \{c_1, \ldots, c_m\}$

**Requires:** scoring function `score`

**Parameters:** number of positive and negative guesses $k_+$ and $k_-$

1 Randomly split canaries $C$ into two equally-sized sets $C_{\text{IN}}$ and $C_{\text{OUT}}$

2 Let $S = \{s_i\}_{i=1}^m$, where $s_i = \begin{cases} 1 & \text{if } c_i \in C_{\text{IN}} \\ -1 & \text{if } c_i \in C_{\text{OUT}} \end{cases}$

3 Train model $w \leftarrow \mathcal{A}(D \cup C_{\text{IN}})$

4 Compute vector of scores $Y = \{\texttt{score}(w, c_i)\}_{i=1}^m$

5 Sort scores in ascending order $Y' \leftarrow \texttt{sort}(Y)$

6 Construct vector of guesses $T = \{t_i\}_{i=1}^m$, where $t_i =$
$$\begin{cases} 1 & \text{if } Y_i \text{ is among the top } k_+ \text{ scores in } Y \text{ (i.e., } Y_i \geq Y'_{m-k_+}) \text{ // } \texttt{guess } c_i \in C_{\text{IN}} \\ -1 & \text{if } Y_i \text{ is among the bottom } k_- \text{ scores in } Y \text{ (i.e., } Y_i \leq Y'_{k_-}) \text{ // } \texttt{guess } c_i \in C_{\text{OUT}} \\ 0 & \text{otherwise // } \texttt{abstain} \end{cases}$$

7 Compute empirical epsilon $\tilde{\varepsilon}$ (i.e., find the largest $\tilde{\varepsilon}$ such that $S$, $T$, $\tau$, and $\delta$ satisfy Theorem 1)

**Output:** $\tilde{\varepsilon}$

---

**Algorithm 2:** Black-box Auditing - One Run (Mahloujifar et al., 2024)

---

**Input:** privacy parameter $\delta$, training algorithm $\mathcal{A}$, dataset $D$, set of $m$ canaries $C = \{c_1, \ldots, c_m\}$

**Requires:** scoring function `score`

**Parameters:** number of guesses $k$

1 Randomly split canaries $C$ into two equally-sized sets $C_{\text{IN}}$ and $C_{\text{OUT}}$

2 Create disjoint canary sets $E = \{e_i\}_{i=1}^{m/2}$ by randomly pairing canaries from $C_{\text{IN}}$ and $C_{\text{OUT}}$ such that $e_i = (c_{i,1}, c_{i,2})$ for $c_{i,1} \in C_{\text{IN}}$ and $c_{i,2} \in C_{\text{OUT}}$ (each canary $c \in C$ appears in **exactly** one set $e_i$)

3 Train model $w \leftarrow \mathcal{A}(D \cup C_{\text{IN}})$

4 Compute vector of scores $Y = \{|\texttt{score}(w, c_{i,1}) - \texttt{score}(w, c_{i,2})|\}_{i=1}^{m/2}$

5 Sort scores in ascending order $Y' \leftarrow \texttt{sort}(Y)$

6 Construct vector of guesses $T = \{t_i\}_{i=1}^{m/2}$, where
$$t_i = \begin{cases} 1 & \text{if } Y_i \text{ is among the top } k \text{ values in } Y \text{ (i.e., } Y_i \geq Y'_{m-k}) \\ & \text{and } \texttt{score}(w, c_{i,1}) > \texttt{score}(w, c_{i,2}) \text{ // } \texttt{guess } c_{i,1} \in C_{\text{IN}} \\ -1 & \text{if } Y_i \text{ is among the top } k \text{ values in } Y \text{ (i.e., } Y_i \geq Y'_{m-k}) \\ & \text{and } \texttt{score}(w, c_{i,1}) \leq \texttt{score}(w, c_{i,2}) \text{ // } \texttt{guess } c_{i,2} \in C_{\text{IN}} \\ 0 & \text{otherwise // } \texttt{abstain} \end{cases}$$

7 Let number of correct guesses $k' = \sum_{i=1}^{m/2} \mathbb{1}\{t_i = 1\}$

8 Compute empirical epsilon $\tilde{\varepsilon}$ (i.e., find the largest $\tilde{\varepsilon}$ whose corresponding $f$-DP function $f$ passes Algorithm 3 for $m$, $k$, $k'$, $\tau$, and $\delta$.)

**Output:** $\tilde{\varepsilon}$

---

We note that, in contrast, previous works have also studied white-box settings. While the exact assumptions made in this setting can vary (Nasr et al., 2021; 2023; Steinke et al., 2023; Koskela & Mohammadi, 2025), it can be characterized as having fewer restrictions (e.g., access to intermediate training iterations or the ability to inject and modify gradients). While auditing in white-box settings generally leads to higher lower bound estimates due to the strength of the auditor, its assumptions are often far less realistic than those made in black-box auditing.

### 2.1.2 ONE-RUN AUDITING

Early works (Jagielski et al., 2020; Tramer et al., 2022; Nasr et al., 2023) design privacy auditing "attacks" that align with the definition of DP, which bounds the difference in outputs on neighboring

---

**Algorithm 3:** Upper bound probability of making correct guesses (Mahloujifar et al., 2024)

---

**Input:** probability threshold $\tau$, functions $f$ and $f^{-1}$, number of guesses $k$, number of correct guesses $k'$, number of samples $m$, alphabet size $s$

**1** $\forall 0 < i < k'$ set $h[i] = 0$, and $r[i] = 0$

**2** Set $r[k'] = \tau \cdot \frac{c}{m}$

**3** Set $h[k'] = \tau \cdot \frac{c'-c}{m}$

**4** **for** $i \in [k'-1, \ldots, 0]$ **do**

**5** $\quad | \quad h[i] = (s-1)f^{-1}(r[i+1])$

**6** $\quad | \quad r[i] = r[i+1] + \frac{i}{k-i} \cdot (h[i] - h[i+1])$

**7** **if** $r[0] + h[0] \geq \frac{k}{m}$ **then**

**8** $\quad | \quad$ Return True (probability of $k'$ correct guesses (out of $k$) is less than $\tau$)

**9** **else**

**10** $\quad | \quad$ Return False (probability of having $k'$ correct guesses (out of $k$) could be more than $\tau$)

---

datasets that differ by one sample. These audits detect the presence (or absence) of an individual sample over hundreds—if not, thousands—of runs of DP-SGD. The auditing procedure then gives a lower bound on $\varepsilon$ based on the true and false positive rates of the membership inference attacks.

While effective, these multi-run auditing procedures are computational expensive. Consequently, Steinke et al. (2023) propose an alternative procedure that requires only *one* training run. Their strategy inserts multiple canary examples and obtains a lower bound based on how well an attacker can guess whether some canary was used in training. While one-run auditing can sacrifice bound tightness, its ability to audit without multiple runs of DP-SGD make it much more efficient and therefore, practical for larger models. In our work, we consider two primary auditing procedures:

**(1)** Steinke et al. (2023) introduce the concept of privacy auditing using one training run. Given some set of canaries $C$, samples are randomly sampled from $C$ with probability $\frac{1}{2}$ and inserted into the training set. Once the model is trained, the auditor guesses which samples in $C$ were or were not included in the training set. The auditor can make any number of guesses or abstain. We present this procedure in Algorithm 1. The final lower bound on $\varepsilon$ is determined using Theorem 1, which is based on the total number of canaries, the number of guesses, and the number of correct guesses.

**Theorem 1** (Analytic result for approximate DP (Steinke et al., 2023))**.** Suppose $\mathcal{A} : \{-1, 1\}^m \rightarrow \{-1, 0, 1\}^m$ satisfy $(\varepsilon, \delta)$-DP. Let $S \in \{-1, 1\}^m$ be uniformly random and $T = \mathcal{A}(S)$. Suppose $\mathbb{P}[\|T\|_1 \leq r] = 1$. Then, for all $v \in \mathbb{R}$,

$$\mathbb{P}_{\substack{S \leftarrow \{-1,1\}^m \\ T \leftarrow M(S)}} \left[ \sum_{i=1}^m \max\{0, T_i \cdot S_i\} \geq v \right] \leq f(v) + 2m\delta \cdot \max_{i \in \{1, \ldots, m\}} \left\{ \frac{f(v-i) - f(v)}{i} \right\},$$

where

$$f(v) := \mathbb{P}_{\tilde{W} \leftarrow \text{Binomial}\left(r, \frac{e^\varepsilon}{e^\varepsilon + 1}\right)} \left[ \tilde{W} \geq v \right].$$

At a very high level, $\mathcal{A}$ is DP-SGD, which takes in as input some set of $m$ canaries that are labeled ($S \in \{-1, 1\}^m$) as being included or excluded from the training set. The auditor uses the output of DP-SGD to produce a vector of guesses $T \in \{-1, 0, 1\}^m$ for the $m$ canaries. Theorem 1 bounds the probability of making at least $v$ correct guesses ($\sum_{i=1}^m \max\{0, T_i \cdot S_i\} \geq v$, where $T_i \cdot S_i = 1$ if the guess is correct). More informally, this theorem bounds the success rate (number of correct guesses) of the auditor assuming the parameter $\varepsilon$. Practically speaking, one runs binary search (Steinke et al., 2023, Appendix D) to estimate the largest $\varepsilon$ such that Theorem 1 still holds.

**(2)** Mahloujifar et al. (2024) propose an alternative auditing procedure that empirically achieves tighter privacy estimates in the white-box setting. In their guessing game, the set of canaries $C$ is randomly partitioned in disjoint sets. One canary is sampled from each set and inserted into the training set. Again, once the model is trained with DP-SGD, the auditor must make guesses. Unlike in Steinke et al. (2023), however, the auditor must guess which canary out of each set was included in training. Algorithm 2 presents this procedure for canary sets of size 2.

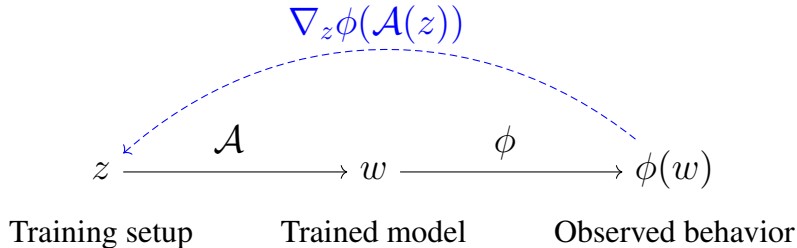

Figure 1: An illustration of the metagradient. We embed the canaries into a continuous metaparameter $z \in \mathbb{R}^{m \times H \times W \times C}$ with one coordinate per training data pixel. All aspects of the learning process other than $z$—the base training data, optimizer hyperparameters, etc.—are baked into the learning algorithm $\mathcal{A}$. The metaparameter thus defines a model $w = \mathcal{A}(z)$, which we use to compute an output metric $\phi(w)$. The metagradient $\nabla_z \phi(\mathcal{A}(z))$ is the gradient of the metric with respect to the metaparameter $z$.

Similar to Steinke et al. (2023), the final lower bound on $\varepsilon$ is determined based on the total number of canary sets, the number of guesses, and the number of correct guesses. At a high level, Mahloujifar et al. (2024) first construct a set of candidate values for $\varepsilon$ and a corresponding $f$-DP function for each. Using Algorithm 3, they then run a hypothesis test, with probability threshold $\tau$, for the number of correct guesses (i.e., output of Algorithm 3) occurring given function $f$. The final empirical lower bound is the maximum $\varepsilon$ among those corresponding to the functions $f$ that pass Algorithm 3.

**Scoring function.** Finally, to determine membership for either procedure, the auditor must first choose some `score` function $s(\cdot)$ from the training process. In the black-box setting for image classification models, one natural choice for $s(\cdot)$ is to use negative cross-entropy loss (Steinke et al., 2023). When $s(w, x)$ is large (i.e., cross-entropy loss is small) for some canary $x$ and model $w$, the auditor guesses that $x$ was included in training, and vice-versa. In Section 4, we provide more details about how we use the score function for Algorithms 1 and 2.

## 3 CANARY OPTIMIZATION WITH METAGRADIENT DESCENT

For a fixed black-box auditing algorithm $\texttt{BBaudit} : (\tau, \delta, \mathcal{A}, D, C) \to \widetilde{\varepsilon}$ (e.g., Algorithm 1 or 2), the main degree of freedom available to the auditor is the choice of canary set $C$. Typically, one chooses $C$ to be a random subset of the training dataset $D$, or a random set of mislabeled examples (Steinke et al., 2023; Mahloujifar et al., 2024). A natural question to ask is whether such choices are (near-)optimal; in other words, *can we significantly improve the efficacy of a given auditing algorithm by carefully designing the canary set $C$?*

In this section, we describe an optimization-based approach to choosing the canary set. At a high level, our goal is to solve an optimization problem of the form

$$\max_C \texttt{BBaudit}(\tau, \delta, \mathcal{A}, D, C), \tag{2}$$

where `BBaudit` is the (fixed) differential privacy auditing algorithm, $\tau$ and $\delta$ are the privacy parameters, $\mathcal{A}$ is the learning algorithm (e.g., DP-SGD), $D$ is the dataset, and $C$ is the set of canary samples. The high-dimensional nature of this problem (e.g., for CIFAR-10, $C \in \mathbb{R}^{m \times 32 \times 32 \times 3}$) makes it impossible to exhaustively search over all possible canary sets $C$.

Instead, the main idea behind our approach is to use *gradient descent* to optimize the canary set $C$. To do so, we first design a surrogate objective function to `audit` by leveraging the connection between membership inference and differential privacy auditing. We then use recent advances in *metagradient* computation (Engstrom et al., 2025) to optimize this surrogate objective with respect to the canary set $C$.

**Key primitive: metagradient descent.** A metagradient is a gradient taken *through* the process of training a machine learning model (Maclaurin et al., 2015; Domke, 2012; Bengio, 2000; Baydin

& Pearlmutter, 2014). Specifically, given a learning algorithm $\mathcal{A}$, a (continuous) design parameter $z$ (e.g., learning rate, weight decay, data weights, etc.), and a loss function $\phi$, the metagradient $\nabla_z \phi(\mathcal{A}(z))$ is the gradient of the final loss $\phi$ with respect to the design parameter $z$ (see Figure 1). For very small-scale learning algorithms (e.g., training shallow neural networks), one can compute metagradients by backpropagating through the entire model training process.

While this approach does not scale directly to larger-scale training routines, the recent work of Engstrom et al. (2025) proposes a method to compute metagradients efficiently and at scale. The method, called REPLAY, enables gradient-based optimization of data importance weights, training hyperparameters, and—most relevant to our setting—poisonous training data that hurts overall model accuracy. To tackle the latter setting, Engstrom et al. (2025) show how to compute metagradients of a model's final test loss with respect to the pixels in its training data. Leveraging this method, we will assume that we can efficiently compute metagradients of any differentiable loss function with respect to the pixels of any number of training data points.

**Surrogate objective function.** Even with the ability to compute metagradients efficiently, the optimization problem in (2) is still challenging to solve in the black-box setting. First, the objective function BBaudit has explicit non-differentiable components (e.g., thresholding). Second, taking the metagradient requires more fine-grained access to the model training process than simply observing the final model outputs.

To address both these challenges, we design a surrogate objective function that approximates the original objective (2), inspired by the connection between black-box privacy auditing and membership inference. In particular, inspecting Algorithms 1 and 2, we observe that in both algorithms, we randomly split the canary set into two sets $C_{\text{IN}}$ and $C_{\text{OUT}}$; trains a model on $D \cup C_{\text{IN}}$; and runs a membership inference attack to distinguish between samples in $C_{\text{IN}}$ and $C_{\text{OUT}}$. Intuitively, a good canary sample $z_i$ should thus satisfy the following properties: (1) **memorizability**: if $z_i \in C_{in}$, the model should have *low* loss on $z_i$ and (2) **non-generalizability**: if $z_i \in C_{out}$, the model should have *high* loss on $z_i$. These properties motivate the following simple surrogate objective function:

$$\phi(w) = \sum_{i=1}^{m} (\mathbf{1}\{z_i \in C_{in}\} - \mathbf{1}\{z_i \in C_{out}\}) \cdot \mathcal{L}(w, z_i), \tag{3}$$

where $\mathcal{L}$ is the training loss (i.e., cross-entropy) and $\mathbf{1}\{\cdot\}$ is the indicator function.[1] Finally, to optimize this objective in the black-box setting, we consider a "standard" (i.e., not differentially private) training algorithm $\mathcal{A}$, and then transfer the resulting canaries to whatever learning algorithm we are auditing.

**Optimizing canaries with (meta-)gradients.** Our final optimization process (given in more detail in Algorithm 4) proceeds in $T > 1$ *metasteps*. Let $D$ be the set of non-canaries (e.g., the CIFAR-10 dataset) and $C$ be the set of canaries (i.e., metaparameters $z$) we are optimizing. During each *metastep* $t$, we randomly partition the canaries $C$ into two sets $C_{\text{IN},t}$ and $C_{\text{OUT},t}$, and randomly sample a model initialization and data ordering which define a learning algorithm $\mathcal{A}$. After training a model $w = \mathcal{A}(z)$, we take a gradient step to minimize the objective (equation 3) with respect to the canary set $C$. By repeating this process several times (essentially running stochastic gradient descent across random seeds and random data orderings partitionings of the canary set), we obtain a set of canary examples that are robustly memorizable and non-generalizable.

## 4 Empirical Evaluation

### 4.1 Setup

**Audited models.** Following prior work (Nasr et al., 2023; Steinke et al., 2023; Mahloujifar et al., 2024), we audit Wide ResNet models (Zagoruyko & Komodakis, 2016) trained on CIFAR-10 (Krizhevsky et al., 2009) with DP-SGD. Like in these works, we specifically use the Wide ResNet

---

[1]We note that in this case, $\phi$ depends on the model weights $w$ but also has a direct dependence on the canary set (i.e., the metaparameters $z$), making Figure 1 a slight over-simplification. In practice, we can still use the law of total derivative to compute the gradient of $\phi$ with respect to $z$, since $\nabla_z \phi(z, \mathcal{A}(z)) = \frac{\partial \phi}{\partial w} \cdot \nabla_z \mathcal{A}(z) + \frac{\partial \phi}{\partial z}$.

---

**Algorithm 4:** Metagradient Canary Optimization

---

**Input:** dataset $D$
**Requires:** training algorithm $\mathcal{A}$, loss function $\mathcal{L}$
**Parameters:** number of canaries $m$, number of meta-iterations $N$
1 Initialize canaries $C_0 = \{c_1, \ldots, c_m\}$
2 **for** $t \leftarrow 0$ **to** $N - 1$ **do**
3   Randomly split $C_t$ into two equally-sized sets: $C_{\text{IN},t}$ and $C_{\text{OUT},t}$
4   Train model: $w_t \leftarrow \mathcal{A}(D \cup C_{\text{IN},t})$
5   Compute loss gap $\phi(w_t) = \mathcal{L}(w_t, C_{\text{IN},t}) - \mathcal{L}(w_t, C_{\text{OUT},t})$
6   Compute gradient w.r.t. canaries: $\nabla_{C_t} \leftarrow \texttt{REPLAY}(w_t, \phi(\theta_t))$
7   Update canaries: $C_{t+1} \leftarrow \text{update}(C_t, \nabla_{C_t})$
**Output:** optimized canaries $C_N$

---

(WRN) 16-4 architecture proposed by De et al. (2022), which they modify to improve model performance after DP training. In addition, we expand our evaluation to include CIFAR-100 (Krizhevsky et al., 2009), MNIST (LeCun et al., 2002), Fashion-MNIST (Xiao et al., 2017). For CIFAR-100, we use the same WRN-16-4 architecture we use for CIFAR-10. Given the simplicity (i.e., that it is greyscale) of MNIST and Fashion-MNIST, we use a simple CNN with 3 convolutional layers, which still achieves high accuracy at $\varepsilon = 8$.

To audit the models, we use canary sets of size $m = 1000$. To remain consistent with Steinke et al. (2023) and Mahloujifar et al. (2024), where $C$ is sampled from the training set, we have in total $r = 49000$ non-canaries training images ($n = 49500$ total training images) for CIFAR-10 and CIFAR-100, and $r = 59000$ ($n = 59500$ total training images) for MNIST and Fashion-MNIST.

Unlike in prior work on one-run auditing, which audits DP-SGD applied to models initialized from scratch, we also expand our evaluation to DP models that are first pretrained nonprivately (i.e., DP-finetuning). For DP-finetuning experiments using CIFAR-10 and CIFAR-100, we take CINIC-10 (Darlow et al., 2018), which combines CIFAR-10 with images from ImageNet (Deng et al., 2009) that correspond to the classes in CIFAR-10. For our pretraining dataset, we remove CIFAR-10 images so that only ImageNet examples remain. For DP-finetuning experiments on MNIST and Fashion-MNIST, we use EMNIST (Cohen et al., 2017), which contains handwritten digits, as well as uppercase and lowercase letters. Except for CIFAR-10, which shares the same classes as its pretraining dataset, all pretrained models' classification layers are replaced before DP training.

We present hyperparameters in Tables 4 and 5 of the appendix. We train our models using the `JAX-Privacy` package (Balle et al., 2025).

**Baselines.** To evaluate our canaries, we follow the black-box experiments of prior work on one-run auditing, comparing against canaries randomly sampled from the training set (Steinke et al., 2023; Mahloujifar et al., 2024), as well as canaries that have been mislabeled (Steinke et al., 2023).

### 4.2 RESULTS

In Tables 1 and 2, we present our main results for both DP training (i.e., training from scratch) and DP finetuning (i.e., first pretraining non-privately). At higher privacy budgets, our method performs better than black-box canaries proposed in prior one-run auditing work (Steinke et al., 2023; Mahloujifar et al., 2024). Under more stringent privacy constraints, in particular for $\varepsilon = 1.0$ (and in some cases, $\varepsilon = 2.0$ for Steinke et al. (2023)'s auditing procedure), the improvements become less consistent. We attribute such variability primarily to the increased noise substantially attenuating memorization signals and thereby limiting the effectiveness of auditing methods overall. Finally, we note that our canaries reinforce that Mahloujifar et al. (2024)'s procedure consistently estimates higher privacy lower bounds compared to that of Steinke et al. (2023), an advantage that was less clear when using standard black-box canaries (i.e., random or mislabeled images). Using Mahloujifar et al. (2024)'s procedure, we estimate lower bounds that are several times higher compared to prior work, regardless of dataset or model initialization.

Table 1: We audit a Wide ResNet 16-4 model that has been trained with DP-SGD ($\varepsilon = \{8.0, 6.0, 4.0, 2.0, 1.0\}, \delta = 10^{-5}$). We present results for models initialized from scratch and trained on private data (DP Training) and pretrained on public data and subsequently privately fine-tuned on the specific dataset (DP Finetuning). We report the average empirical epsilon over 5 runs for auditing procedure introduced by Steinke et al. (2023).

| Dataset | Training | Canary Type | $\varepsilon = 8$ | $\varepsilon = 6$ | $\varepsilon = 4$ | $\varepsilon = 2$ | $\varepsilon = 1$ |
|---|---|---|---|---|---|---|---|
| CIFAR-10 | DP Training | Metagradient (*ours*) | **0.392** | **0.271** | **0.312** | **0.146** | **0.098** |
| | | Random | 0.285 | 0.111 | 0.044 | 0.083 | 0.080 |
| | | Random Mislabeled | 0.308 | 0.150 | 0.088 | 0.017 | 0.018 |
| | DP Finetuning | Metagradient (*ours*) | **0.665** | **0.584** | 0.315 | 0.157 | **0.153** |
| | | Random | 0.477 | 0.322 | **0.323** | **0.200** | 0.065 |
| | | Random Mislabeled | 0.489 | 0.246 | 0.251 | 0.178 | 0.063 |
| CIFAR-100 | DP Training | Metagradient (*ours*) | **0.422** | **0.410** | 0.141 | 0.069 | 0.074 |
| | | Random | 0.148 | 0.134 | **0.170** | 0.131 | 0.075 |
| | | Random Mislabeled | 0.165 | 0.194 | 0.140 | 0.126 | **0.143** |
| | DP Finetuning | Metagradient (*ours*) | **0.770** | **0.555** | **0.405** | 0.171 | **0.169** |
| | | Random | 0.342 | 0.231 | 0.100 | 0.091 | 0.085 |
| | | Random Mislabeled | 0.175 | 0.200 | 0.147 | 0.150 | 0.165 |
| MNIST | DP Training | Metagradient (*ours*) | **0.392** | **0.438** | **0.280** | 0.147 | 0.150 |
| | | Random | 0.212 | 0.200 | 0.195 | 0.214 | 0.217 |
| | | Random Mislabeled | 0.222 | 0.352 | 0.235 | **0.222** | **0.248** |
| | DP Finetuning | Metagradient (*ours*) | **0.736** | **0.601** | **0.650** | **0.364** | 0.142 |
| | | Random | 0.254 | 0.239 | 0.249 | 0.230 | 0.214 |
| | | Random Mislabeled | 0.366 | 0.319 | 0.425 | 0.304 | **0.227** |
| Fashion-MNIST | DP Training | Metagradient (*ours*) | **0.358** | **0.419** | **0.480** | **0.246** | 0.085 |
| | | Random | 0.124 | 0.125 | 0.143 | 0.149 | 0.120 |
| | | Random Mislabeled | 0.222 | 0.177 | 0.195 | 0.186 | **0.240** |
| | DP Finetuning | Metagradient (*ours*) | **0.817** | **0.996** | **0.672** | **0.390** | 0.172 |
| | | Random | 0.268 | 0.257 | 0.266 | 0.225 | **0.218** |
| | | Random Mislabeled | 0.124 | 0.136 | 0.114 | 0.129 | 0.155 |

### 4.2.1 COMPARISON AGAINST NEWLY PROPOSED CANARIES SUGGESTED DURING THE REVIEW PERIOD

In this subsection, we address suggestions made by reviewers for comparing against other canaries not found in the single-run auditing literature. We provide more details about the construction of these canaries in Appendix C and the empirical results in Tables 6 and 7.

First, Reviewer Vbyr suggests comparing against ClipBKD (Jagielski et al., 2020), which is designed for constructing a single canary example used for multi-run privacy auditing. We find, however, that ClipBKD constructs canaries that are ineffective for single-run auditing, which is unsurprising given that it was not originally designed for constructing many canaries required for one-run auditing.

Second, Reviewer qvdv suggests a new method (which we denote as Gradient Max), that takes some surrogate model and optimizes images that maximize the gradient norm of the loss with respect to the surrogate. Unlike ClipBKD, we find that this method is highly effective compared to existing canaries. Compared to our metagradient canaries, results are mixed, with it outperforming ours on CIFAR-10, performing similarly on CIFAR-100, and performing worse on MNIST and Fashion-MNIST. We believe this is promising direction and hope to explore in the future how this objective can be improved or whether it should be integrated in some way into our metagradient optimization pipeline to further close the gap between theoretical and empirical privacy bounds for black-box auditing.

Table 2: We audit a Wide ResNet 16-4 model that has been trained with DP-SGD ($\varepsilon = \{8.0, 6.0, 4.0, 2.0, 1.0\}, \delta = 10^{-5}$). We present results for models initialized from scratch and trained on private data (DP Training) and pretrained on public data and subsequently privately fine-tuned on the specific dataset (DP Finetuning). We report the average empirical epsilon over 5 runs for auditing procedure introduced by Mahloujifar et al. (2024).

| Dataset | Training | Canary Type | $\varepsilon = 8$ | $\varepsilon = 6$ | $\varepsilon = 4$ | $\varepsilon = 2$ | $\varepsilon = 1$ |
|---|---|---|---|---|---|---|---|
| CIFAR-10 | DP Training | Metagradient (*ours*) | **0.732** | **0.498** | **0.229** | **0.049** | 0.000 |
| | | Random | 0.405 | 0.213 | 0.077 | 0.000 | **0.030** |
| | | Random Mislabeled | 0.225 | 0.086 | 0.019 | 0.025 | 0.000 |
| | DP Finetuning | Metagradient (*ours*) | **1.207** | **0.978** | **0.594** | **0.398** | **0.118** |
| | | Random | 0.687 | 0.482 | 0.456 | 0.091 | 0.077 |
| | | Random Mislabeled | 0.632 | 0.618 | 0.378 | 0.307 | 0.016 |
| CIFAR-100 | DP Training | Metagradient (*ours*) | **0.736** | **0.573** | **0.337** | **0.120** | **0.083** |
| | | Random | 0.209 | 0.030 | 0.013 | 0.064 | 0.002 |
| | | Random Mislabeled | 0.088 | 0.097 | 0.054 | 0.074 | 0.029 |
| | DP Finetuning | Metagradient (*ours*) | **1.286** | **0.935** | **0.594** | **0.370** | 0.113 |
| | | Random | 0.354 | 0.274 | 0.065 | 0.115 | 0.038 |
| | | Random Mislabeled | 0.187 | 0.123 | 0.017 | 0.085 | **0.115** |
| MNIST | DP Training | Metagradient (*ours*) | **0.669** | **0.594** | **0.499** | **0.144** | **0.080** |
| | | Random | 0.016 | 0.000 | 0.000 | 0.000 | 0.000 |
| | | Random Mislabeled | 0.070 | 0.204 | 0.142 | 0.039 | 0.008 |
| | DP Finetuning | Metagradient (*ours*) | **1.465** | **1.366** | **1.260** | **0.304** | **0.117** |
| | | Random | 0.321 | 0.182 | 0.229 | 0.098 | 0.058 |
| | | Random Mislabeled | 0.099 | 0.011 | 0.136 | 0.008 | 0.112 |
| Fashion-MNIST | DP Training | Metagradient (*ours*) | **0.682** | **0.632** | **0.642** | **0.385** | 0.102 |
| | | Random | 0.022 | 0.099 | 0.045 | 0.009 | 0.000 |
| | | Random Mislabeled | 0.141 | 0.085 | 0.094 | 0.061 | **0.136** |
| | DP Finetuning | Metagradient (*ours*) | **1.483** | **1.342** | **1.401** | **0.517** | 0.136 |
| | | Random | 0.056 | 0.093 | 0.058 | 0.179 | 0.002 |
| | | Random Mislabeled | 0.398 | 0.198 | 0.242 | 0.062 | **0.235** |

## 5 CONCLUSION

We propose an efficient method for canary optimization that leverages metagradient descent. Optimizing for an objective tailored towards privacy auditing, our canaries significantly outperform standard canaries, which are sampled from the training dataset. Specifically, we show that despite being optimized for non-private SGD on a small ResNet model, our canaries work better on larger Wide ResNets for both DP-training and DP-finetuning. Using our canaries, we significantly tighten privacy bounds. Furthermore, this improvement holds, regardless of whether DP-SGD is run from scratch or after non-private finetuning.

## 6 REPRODUCIBILITY STATEMENT

In our paper, we describe in detail the algorithms for both our metagradient optimization procedure (Algorithm 4) and the one-run auditing procedures from Steinke et al. (2023) (Algorithm 1) and Mahloujifar et al. (2024) (Algorithms 2 and 3). Additional implementation details and hyperparameters for running these algorithms are described in Appendix B. To ensure reproducibility of our results, we also provide code used for all our experiments in the supplementary materials. There are two main parts: code for metagradient canary optimization and code for training and auditing image classification models with DP-SGD. Instructions for running the code are available in an included README file. In the final version of our paper, we will include a link to a public repository.

## 7 ACKNOWLEDGMENTS

ZSW and TL are supported in part by an NSF CAREER Award, an NSF SaTC grant, and an STTR grant. ZSW and AI are supported by a CyLab Seed Grant.

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

# A    ADDITIONAL METHODOLOGY DETAILS

**Differentially Private SGD.**    For completeness, we describe DP-SGD in Algorithm 5.

---

**Algorithm 5:** Differentially Private Stochastic Gradient Descent (DP-SGD)

---

**Input:** $x \in \mathcal{X}^n$
**Requires:** Loss function $f : \mathbb{R}^d \times \mathcal{X} \to \mathbb{R}$
**Parameters:** Number of iterations $\ell$, learning rate $\eta$, clipping threshold $c > 0$, noise multiplier
$\sigma > 0$, sampling probability $q \in (0, 1]$

**1** Initialize $w_0 \in \mathbb{R}^d$;
**2 for** $t = 1, \ldots, \ell$ **do**
**3** $\quad$ Sample $S^t \subseteq [n]$ where each $i \in [n]$ is included independently with probability $q$;
**4** $\quad$ Compute $g_i^t = \nabla_{w^{t-1}} f(w^{t-1}, x_i) \in \mathbb{R}^d$ for all $i \in S^t$;
**5** $\quad$ Clip $\tilde{g}_i^t = \min \left\{ 1, \frac{c}{\|g_i^t\|_2} \right\} \cdot g_i^t \in \mathbb{R}^d$ for all $i \in S^t$;
**6** $\quad$ Sample $\xi^t \in \mathbb{R}^d$ from $\mathcal{N}(0, \sigma^2 c^2 I)$;
**7** $\quad$ Sum $\tilde{g}^t = \xi^t + \sum_{i \in S^t} \tilde{g}_i^t \in \mathbb{R}^d$;
**8** $\quad$ Update $w^t = w^{t-1} - \eta \cdot \tilde{g}^t \in \mathbb{R}^d$;
**Output:** $w^0, w^1, \ldots, w^\ell$

---

**Additional auditing procedure details.**    As implemented in Mahloujifar et al. (2024), we align Algorithms 1 and 2 by fixing the canary set size to 2 so that half of $C$ is included in training for both auditing setups. When running Algorithm 1, we split $C$ randomly in half (instead of sampling with probability half) so that the set of $r$ non-auditing examples are the same for both auditing procedures. In addition, we use negative cross-entropy loss as the scoring function $s(\cdot)$ for both algorithms. In more detail,

- **[Steinke et al. (2023), Algorithm 1]** We sort the canaries $x \in C$ by $s(x)$ and take the top $k_+$ canaries in the sorted list as positive guesses and bottom $k_-$ as negative guesses.
- **[Mahloujifar et al. (2024), Algorithm 2]** We score canaries in each pair and predict the one with the higher score to have been included in training. We then score each pairing by taking the absolute difference scores $s(\cdot)$ between the canaries in each set and ranking the pairs by the difference. We take the top $k$ sets as our guesses.

For both procedures, we follow prior work (Steinke et al., 2023; Mahloujifar et al., 2024), varying the number of guesses from 10 up to $m$, in increments of 10, and reporting the max empirical $\varepsilon$.

## B  ADDITIONAL EXPERIMENTAL DETAILS

**Metagradient canary optimization.**  Following Engstrom et al. (2025), we optimize the canary samples by training a ResNet-9 model (i.e., $w$ in Algorithm 4), allowing us to optimize $C$ efficiently. For step 1 of Algorithm 4, we initialize $C_0$ to $m$ samples randomly sampled from $D$ (i.e., CIFAR-10). We optimize for 500 metagradient steps. As demonstrated in Section 4.2, despite using a relatively compact model, our metagradient canaries are effective for much larger model architectures (i.e., Wide ResNets). In Table 3, we list the hyperparameters used to train the proxy ResNet-9 model at each metagradient step. While our metagradient optimization approach introduces a non-trivial computational overhead, we note that this expense is very practical when considering it only needs to be performed **once** per dataset. In fact, once optimized for a specific dataset, these canaries can be reused across various model architectures, different privacy budgets ($\epsilon$), and diverse training procedures (e.g., training from scratch versus DP-finetuning), effectively amortizing the initial optimization cost. Specifically, optimizing a full set of $m = 1000$ canaries requires approximately 96–120 hours on a single NVIDIA A100 GPU.

Table 3: ResNet-9 Training Hyperparameters

| Hyperparameter | Value |
|---|---|
| Learning rate | 0.2 |
| $\beta_1$ | 0.85 |
| Weight Decay | $1 \times 10^{-5}$ |
| Batch size | 250 |
| Epochs | 18 |
| Optimizer | SGD |
| Nesterov | True |
| Momentum | False |

**DP-SGD Hyperparameters.**  We list the hyperparameters used for DP-SGD in Tables 4 and 5. CIFAR-10 (training from scratch) hyperparameters were directly taken from De et al. (2022). We chose the remaining hyperparameters based on a light gridsearch that gave the best test accuracy at each privacy budget.

Table 4: Hyperparameters used for DP training (from scratch).

| | Hyperparameter | $\varepsilon = 8.0$ | $\varepsilon = 6.0$ | $\varepsilon = 4.0$ | $\varepsilon = 2.0$ | $\varepsilon = 1.0$ |
|---|---|---|---|---|---|---|
| CIFAR-10 | Batch size | 4096 | 4096 | 4096 | 4096 | 4096 |
| | Clipping norm | 1.0 | 1.0 | 1.0 | 1.0 | 1.0 |
| | # steps | 2500 | 2000 | 1500 | 1250 | 1000 |
| | Learning rate | 4.0 | 4.0 | 2.0 | 2.0 | 2.0 |
| CIFAR-100 | Batch size | 4096 | 4096 | 4096 | 4096 | 4096 |
| | Clipping norm | 1.0 | 1.0 | 1.0 | 1.0 | 1.0 |
| | # steps | 2500 | 2000 | 1500 | 1250 | 1000 |
| | Learning rate | 4.0 | 4.0 | 2.0 | 2.0 | 2.0 |
| MNIST | Batch size | 4096 | 4096 | 4096 | 4096 | 4096 |
| | Clipping norm | 1.0 | 1.0 | 1.0 | 1.0 | 1.0 |
| | # steps | 1000 | 1000 | 1000 | 1000 | 1000 |
| | Learning rate | 4.0 | 4.0 | 4.0 | 4.0 | 4.0 |
| Fashion-MNIST | Batch size | 4096 | 4096 | 4096 | 4096 | 4096 |
| | Clipping norm | 1.0 | 1.0 | 1.0 | 1.0 | 1.0 |
| | # steps | 1000 | 1000 | 1000 | 1000 | 1000 |
| | Learning rate | 4.0 | 4.0 | 4.0 | 4.0 | 4.0 |

Table 5: Hyperparameters used for DP finetuning.

|  | **Hyperparameter** | $\varepsilon = 8.0$ | $\varepsilon = 6.0$ | $\varepsilon = 4.0$ | $\varepsilon = 2.0$ | $\varepsilon = 1.0$ |
|---|---|---|---|---|---|---|
| CIFAR-10 | Batch size | 4096 | 4096 | 4096 | 4096 | 4096 |
|  | Clipping norm | 1.0 | 1.0 | 1.0 | 1.0 | 1.0 |
|  | # steps | 1000 | 1000 | 740 | 500 | 250 |
|  | Learning rate | 4.0 | 4.0 | 4.0 | 4.0 | 4.0 |
| CIFAR-100 | Batch size | 4096 | 4096 | 4096 | 4096 | 4096 |
|  | Clipping norm | 1.0 | 1.0 | 1.0 | 1.0 | 1.0 |
|  | # steps | 2500 | 2000 | 1500 | 1250 | 1000 |
|  | Learning rate | 4.0 | 4.0 | 2.0 | 2.0 | 2.0 |
| MNIST | Batch size | 4096 | 4096 | 4096 | 4096 | 4096 |
|  | Clipping norm | 1.0 | 1.0 | 1.0 | 1.0 | 1.0 |
|  | # steps | 1000 | 1000 | 1000 | 1000 | 1000 |
|  | Learning rate | 4.0 | 4.0 | 4.0 | 4.0 | 4.0 |
| Fashion-MNIST | Batch size | 4096 | 4096 | 4096 | 4096 | 4096 |
|  | Clipping norm | 1.0 | 1.0 | 1.0 | 1.0 | 1.0 |
|  | # steps | 1000 | 1000 | 1000 | 1000 | 1000 |
|  | Learning rate | 4.0 | 4.0 | 4.0 | 4.0 | 4.0 |

## C   COMPARING AGAINST NEWLY PROPOSED CANARIES

**ClipBKD (Jagielski et al., 2020).**   ClipBKD (Jagielski et al., 2020) is a method for generating canaries that was originally designed for multi-run DP auditing and subsequently used to audit worst-cast privacy leakage (Muthu Selva Annamalai & De Cristofaro, 2024). The method requires taking the top singular principal component of the dataset, and so to modify it for the one-run setting, we instead take the top $m$ components to generate $m$ canaries. Given that for MNIST and Fashion-MNIST, the dimension-size of the images is $d = 784 < m = 1000$, we evaluate ClipBKD on CIFAR-10 and CIFAR-100 only.

**Gradient Maximization (proposed by Reviewer qvdv).**   Reviewer qvdv suggested trying a simple baseline of maximizing the gradient norm of the loss for some surrogate model trained non-privately on the dataset. We refer to this method as Gradient Max. To the best of our knowledge, no previous work has adopted such a canary optimization method in the context of DP auditing. To make this baseline comparable to our metagradient optimization procedure, we also choose ResNet-9 as the surrogate model that we pretrain non-privately on each target dataset. We then randomly sample 1000 holdout samples from these datasets' test sets and optimize the images using gradient descent to maximize the gradient norm of cross-entropy loss.

Table 6: We present the results of a comparison between our method and the baseline proposed by the reviewers, as discussed in Appendix C. We audit a Wide ResNet 16-4 model that has been trained with DP-SGD ($\varepsilon = \{8.0, 6.0, 4.0, 2.0, 1.0\}, \delta = 10^{-5}$). We present results for models initialized from scratch and trained on private data (DP Training) and pretrained on public data and subsequently privately finetuned on the specific dataset (DP Finetuning). We report the average empirical epsilon over 5 runs for auditing procedure introduced by Steinke et al. (2023)

| Dataset | Training | Canary Type | $\varepsilon = 8$ | $\varepsilon = 6$ | $\varepsilon = 4$ | $\varepsilon = 2$ | $\varepsilon = 1$ |
|---|---|---|---|---|---|---|---|
| CIFAR-10 | DP Training | Metagradient (*ours*) | 0.392 | 0.271 | **0.312** | **0.146** | 0.098 |
| | | ClipBKD | 0.129 | 0.162 | 0.116 | 0.134 | 0.094 |
| | | Gradient Max | **0.589** | **0.499** | 0.262 | 0.046 | **0.237** |
| | DP Finetuning | Metagradient (*ours*) | 0.665 | 0.584 | **0.315** | 0.157 | **0.153** |
| | | ClipBKD | 0.487 | 0.282 | 0.160 | 0.151 | 0.041 |
| | | Gradient Max | **0.909** | **0.738** | 0.186 | **0.411** | 0.111 |
| CIFAR-100 | DP Training | Metagradient (*ours*) | **0.422** | **0.410** | 0.141 | 0.069 | **0.074** |
| | | ClipBKD | 0.219 | 0.220 | 0.087 | **0.226** | 0.066 |
| | | Gradient Max | 0.378 | 0.303 | **0.280** | 0.125 | **0.074** |
| | DP Finetuning | Metagradient (*ours*) | **0.770** | 0.555 | **0.405** | 0.171 | 0.169 |
| | | ClipBKD | 0.213 | 0.174 | 0.200 | 0.192 | 0.129 |
| | | Gradient Max | 0.623 | **0.579** | 0.321 | **0.258** | **0.265** |
| MNIST | DP Training | Metagradient (*ours*) | 0.392 | **0.438** | **0.280** | 0.147 | **0.150** |
| | | Gradient Max | **0.442** | 0.272 | 0.249 | **0.151** | 0.108 |
| | DP Finetuning | Metagradient (*ours*) | **0.736** | **0.601** | **0.650** | **0.364** | 0.142 |
| | | Gradient Max | 0.539 | 0.454 | 0.568 | 0.260 | **0.170** |
| Fashion-MNIST | DP Training | Metagradient (*ours*) | **0.358** | **0.419** | **0.480** | **0.246** | **0.085** |
| | | Gradient Max | 0.277 | 0.272 | 0.202 | 0.230 | 0.061 |
| | DP Finetuning | Metagradient (*ours*) | **0.817** | **0.996** | **0.672** | **0.390** | **0.172** |
| | | Gradient Max | 0.289 | 0.158 | 0.123 | 0.202 | 0.102 |

Table 7: We present the results of a comparison between our method and the baseline proposed by the reviewers, as discussed in Appendix C. We audit a Wide ResNet 16-4 model that has been trained with DP-SGD ($\varepsilon = \{8.0, 6.0, 4.0, 2.0, 1.0\}, \delta = 10^{-5}$). We present results for models initialized from scratch and trained on private data (DP Training) and pretrained on public data and subsequently privately finetuned on the specific dataset (DP Finetuning). We report the average empirical epsilon over 5 runs for auditing procedure introduced by Mahloujifar et al. (2024).

| Dataset | Training | Canary Type | $\varepsilon = 8$ | $\varepsilon = 6$ | $\varepsilon = 4$ | $\varepsilon = 2$ | $\varepsilon = 1$ |
|---|---|---|---|---|---|---|---|
| CIFAR-10 | DP Training | Metagradient (*ours*) | 0.732 | 0.498 | 0.229 | 0.049 | 0.000 |
| | | ClipBKD | 0.270 | 0.243 | 0.193 | **0.112** | 0.165 |
| | | Gradient Max | **0.970** | **0.595** | **0.272** | 0.070 | **0.263** |
| | DP Finetuning | Metagradient (*ours*) | 1.207 | 0.978 | **0.594** | 0.398 | 0.118 |
| | | ClipBKD | 0.841 | 0.344 | 0.184 | 0.187 | 0.030 |
| | | Gradient Max | **1.793** | **1.249** | 0.403 | **0.409** | **0.368** |
| CIFAR-100 | DP Training | Metagradient (*ours*) | 0.736 | **0.573** | 0.337 | **0.120** | 0.083 |
| | | ClipBKD | 0.248 | 0.243 | 0.150 | 0.077 | 0.025 |
| | | Gradient Max | **0.757** | 0.509 | **0.414** | 0.102 | **0.099** |
| | DP Finetuning | Metagradient (*ours*) | 1.286 | 0.935 | 0.594 | 0.370 | 0.113 |
| | | ClipBKD | 0.238 | 0.371 | 0.206 | 0.126 | **0.177** |
| | | Gradient Max | **1.303** | **1.053** | **0.658** | **0.538** | 0.103 |
| MNIST | DP Training | Metagradient (*ours*) | **0.669** | **0.594** | **0.499** | **0.144** | 0.080 |
| | | Gradient Max | 0.485 | 0.403 | 0.339 | 0.089 | **0.083** |
| | DP Finetuning | Metagradient (*ours*) | **1.465** | **1.366** | **1.260** | 0.304 | 0.117 |
| | | Gradient Max | 0.924 | 0.884 | 0.823 | **0.451** | **0.147** |
| Fashion-MNIST | DP Training | Metagradient (*ours*) | **0.682** | **0.632** | **0.642** | **0.385** | **0.102** |
| | | Gradient Max | 0.453 | 0.451 | 0.320 | 0.297 | 0.026 |
| | DP Finetuning | Metagradient (*ours*) | **1.483** | **1.342** | **1.401** | **0.517** | **0.136** |
| | | Gradient Max | 0.227 | 0.256 | 0.227 | 0.329 | 0.064 |

# D  ADDITIONAL ANALYSIS USING CIFAR-10

**Auditing non-private SGD.**   We evaluate our metagradient canaries when auditing models trained with *non-private* SGD. In Figure 2, we plot the empirical epsilon estimated by the auditing procedures introduced in Steinke et al. (2023) and Mahloujifar et al. (2024) against the number of steps that the Wide ResNet 16-4 model is trained for. We observe that even when applied on different model architectures (i.e., transferring from ResNet-9 to WRN 16-4), our metagradient canaries perform strongly. Using the auditing procedure of Mahloujifar et al. (2024), for example, our canaries outperform the two baselines across any number of training steps.

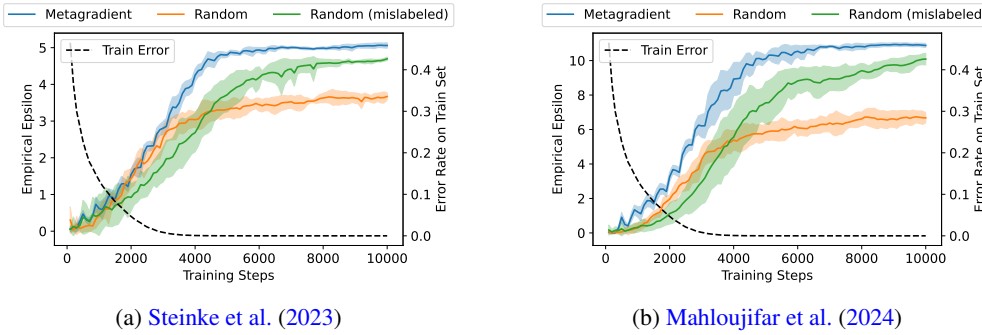

(a) Steinke et al. (2023)          (b) Mahloujifar et al. (2024)

Figure 2: We evaluate the effectiveness of our metagradient canaries for the purpose of auditing *non-private* SGD. We train a Wide ResNet 16-4 model on CIFAR-10 for $10k$ steps with each canary type, plotting the empirical epsilon when auditing the model at every $100$ steps with the auditing procedures introduced by **(a)** Steinke et al. (2023) and **(b)** Mahloujifar et al. (2024). We take an average over 5 runs and plot an error band to denote $\pm 1$ standard deviation. For reference, we plot the training error of the model trained on our metagradient canaries (note that the training accuracy is approximately the same, regardless of choice of canary).

**Canaries visualization.**   To provide insights into the effectiveness of the optimized canary sets, we include visualizations of the generated images by our procedure. To the human eye, the optimized canaries appear as diverse, random shapes and patterns. This suggests that our metagradient optimization identifies specific features that might be susceptible to memorization by the model, even if those features have no semantic meaning to a human observer.

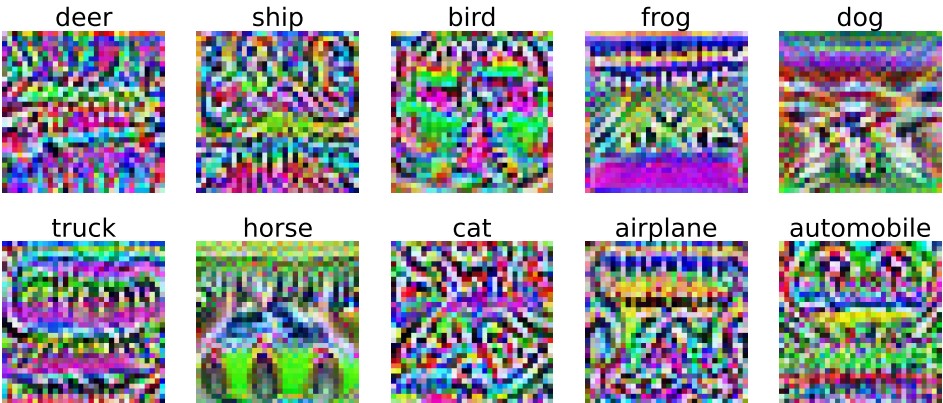

Figure 3: CIFAR10 Metagradient Canaries - Generated using Algorithm 4

**Evaluation on vision transformers.**   In Tables 8 and 9, we present auditing results for vision transformer models (i.e., TinyViT (Wu et al., 2022)) trained with DP-SGD at $\varepsilon = 8.0$ using the auditing procedures of Steinke et al. (2023) and Mahloujifar et al. (2024) respectively. In these tables, we also compared the ClipBKD and Gradient Max canaries that were described in Appendix C.

Table 8: Using the procedure presented in Steinke et al. (2023), we audit a ViT-Tiny model that has been trained with DP-SGD ($\varepsilon = 8.0$, $\delta = 10^{-5}$) on CIFAR-10 and CIFAR-100. We present results for models initialized from scratch and pretrained on CINIC-10 (with CIFAR-10 images removed). We report the average empirical epsilon and standard error over 5 runs.

| Dataset | Training | Canary Type | $\varepsilon = 8$ |
|---|---|---|---|
| CIFAR-10 | DP Training | Metagradient (*ours*) | **0.420** |
| | | Random | 0.088 |
| | | Random Mislabeled | 0.255 |
| | | ClipBKD | 0.192 |
| | | Gradient Max | 0.392 |
| | DP Finetuning | Metagradient (*ours*) | **0.943** |
| | | Random | 0.164 |
| | | Random Mislabeled | 0.271 |
| | | ClipBKD | 0.114 |
| | | Gradient Max | 0.146 |
| CIFAR-100 | DP Training | Metagradient (*ours*) | 0.560 |
| | | Random | 0.167 |
| | | Random Mislabeled | 0.020 |
| | | ClipBKD | 0.433 |
| | | Gradient Max | **0.667** |
| | DP Finetuning | Metagradient (*ours*) | **1.210** |
| | | Random | 0.279 |
| | | Random Mislabeled | 0.147 |
| | | ClipBKD | 0.248 |
| | | Gradient Max | 0.540 |

Table 9: Using the procedure presented in Mahloujifar et al. (2024), we audit a ViT-Tiny model that has been trained with DP-SGD ($\varepsilon = 8.0$, $\delta = 10^{-5}$) on CIFAR-10 and CIFAR-100. We present results for models initialized from scratch and pretrained on CINIC-10 (with CIFAR-10 images removed). We report the average empirical epsilon and standard error over 5 runs.

| Dataset | Training | Canary Type | $\varepsilon = 8$ |
|---|---|---|---|
| CIFAR-10 | DP Training | Metagradient (*ours*) | **0.712** |
| | | Random | 0.093 |
| | | Random Mislabeled | 0.176 |
| | | ClipBKD | 0.069 |
| | | Gradient Max | 0.502 |
| | DP Finetuning | Metagradient (*ours*) | **1.742** |
| | | Random | 0.288 |
| | | Random Mislabeled | 0.419 |
| | | ClipBKD | 0.214 |
| | | Gradient Max | 0.205 |
| CIFAR-100 | DP Training | Metagradient (*ours*) | 0.868 |
| | | Random | 0.173 |
| | | Random Mislabeled | 0.064 |
| | | ClipBKD | 0.235 |
| | | Gradient Max | **0.932** |
| | DP Finetuning | Metagradient (*ours*) | **1.954** |
| | | Random | 0.397 |
| | | Random Mislabeled | 0.091 |
| | | ClipBKD | 0.344 |
| | | Gradient Max | 1.081 |

