# OpenReview forum: "Optimizing Canaries for Privacy Auditing with Metagradient Descent"
_ICLR.cc/2026/Conference — ICLR 2026 Poster_

### Official Review · Reviewer_Vbyr · 2025-10-25

**Soundness:** 3
**Presentation:** 3
**Contribution:** 2
**Rating:** 4
**Confidence:** 4

**Summary:**

Differential privacy (DP) is a privacy measure of a mechanism - a randomized function from a set of data points to some output. Its privacy is measured by the ability of an adversary to distinguish between two possible input datasets differing in a single element (referred to as neighbors), based on a single output of the mechanism. The better the privacy, the close is the success rate of the most sophisticated analyst to a random guess.
This framing as an hypothesis testing naturally lends itself to an empirical lower bound scheme; select two neighboring datasets, repeatedly run the mechanism on one one of the two datasets selected at random, and guess the identity of the used dataset based on the mechanism's output. The more accurate the guesser, the lower the privacy. Recently, a more efficient method has emerged, where rather than repeating many time the experiment to determine a change in a single element, the mechanism is called once and the auditing attempts to identify the participation of many elements, but the general idea remains.

Naturally, such auditing can only provide a lower bound, since DP is a worst-case property, bounding the privacy loss of *any* neighboring datasets against *any* guesser. As a result, a tight lower bounds requires selecting "easy to distinguish" elements (referred to as canaries), which increase the success rate of the guesser. This work considers a new method for curating such canaries using meta-gradient optimization, where a hyper parameter of the optimization process (e.g., learning rate) is optimized via GD, by propagating the gradient through the entire learning process. The authors propose a somewhat simplified attack which is differentiable, and show that using it top curate gradients improves the auditing results.

**Strengths:**

* This work addresses a known challenge, achieving a tighter lower bound in DP auditing by optimizing the choice of canary(es).
* It achieves this goal using the natural and novel (in this context, to the best of my knowledge) method of meta-gradients.
* The authors empirically back their idea by several experiments.
* The presentation is clear and detailed.

**Weaknesses:**

While the idea of using meta-gradients for canaries optimization seems promising, the implementation considered in this work seems over simplified, and the empirical comparison does not support the claims of meaningful improvement. Concretely, the Surrogate objective function (Eq. 3) captures a naive auditing method which relies only of the change in loss, which is one of the weakest auditing techniques even in the context of black-box auditing. Furthermore, the authors only compare their results to two of the weakest methods for canaries curation - random choice or elements from the underlying distribution or random elements with random labels. The existing literature contains many advanced canary curation methods, some of which were mentioned in the related work section, and without proper comparison it is impossible to asses the contribution of this work.

**Questions:**

Can the authors address my main concern, lack of comparison to other canary curation methods discussed in the literature, such as the ones mentioned by the authors and in [1] (the black-box case)?

[1] Nasr, Milad, et al. "Adversary instantiation: Lower bounds for differentially private machine learning." 2021 IEEE Symposium on security and privacy (SP). IEEE, 2021.

---

> ### Author Response · Authors · 2025-11-19
> **Review Response**
>
> We thank the reviewer for their thoughtful and detailed feedback. Below, we address the main concern regarding the practical applicability of different approaches. Please let us know if there is any additional information we can provide that may influence your decision to update your final score.
> We first would like to point out that what [1] refers to as black-box auditing is exactly the same procedure as the one we use. Similarly, follow-up works [2] and [3], also take black-box auditing to mean the same (i.e., when one can only access the change in loss for auditing).
> In IV.B and C of [1], they discuss auditing methods that are stronger than the black-box setting. While we agree that it is theoretically possible to include such methods, there exist some fundamental differences between [1] and works on one-run auditing (e.g., ours, [3], and [4]) that make these baselines unsuitable. First, IV.B and C are designed for multi-run auditing, which only requires constructing a single canary, rather than a set (of size m=1000 in our work). Moreover, [1] requires the adversary to train shadow models using the same hyperparameters as the ones used for training the audited DP model. While training multiple DP models makes sense for multi-run auditing, it goes against the primary motivation behind one-run auditing (i.e., alleviating the need to run DP-SGD hundreds of times, which using an A100 would require 12 hours for each run). As a result, prior works on one-run auditing only consider the random and mislabeled canaries in their experiments as well.
>     Consequently, we respectfully disagree that the “existing literature contains many advanced canary curation methods, some of which were mentioned in the related work section.” To the best of our knowledge, there does not exist other canary curation baselines suitable for the black-box, last-iterate one-run framework, but we are open to considering other suggestions for curating sets of canaries that are more appropriate for our experiments. Still, we hope that you can consider improving your score, given that our method significantly improves lower bounds compared to prior work [3, 4] on a now expanded set of experiments (see general comments).
>
> [1] Nasr, Milad, et al. "Adversary instantiation: Lower bounds for differentially private machine learning." 2021 IEEE Symposium on security and privacy (SP). IEEE, 2021.
>
> [2] Nasr, Milad, et al. "Tight auditing of differentially private machine learning." 32nd USENIX Security Symposium (USENIX Security 23). 2023.
>
> [3] Steinke, Thomas, Milad Nasr, and Matthew Jagielski. "Privacy auditing with one (1) training run." Advances in Neural Information Processing Systems 36 (2023): 49268-49280.
>
> [4] Mahloujifar, Saeed, Luca Melis, and Kamalika Chaudhuri. "Auditing $ f $-differential privacy in one run." Forty-second International Conference on Machine Learning.

---

> > ### Comment · Reviewer_Vbyr · 2025-11-19
> >
> > To the best of my understanding, the threat model considered in this work actually corresponds to the one considered in Section IV.B in [1], since meta gradients require access to the final model, not just API access. As such, many canary curation methods of samples (rather than gradients) still apply. For example, methods 3 and 4 in Appendix C.3 in [2].
> >
> > Indeed, most referenced works on canaries curation used multiple run auditing framework, as they were published prior to the publication of the one run auditing method. This does not mean these techniques cannot be adapted to the one run setting, by producing many canaries that are much easier to audit. For example, ClipBKD [5] which computes PCA of the data and selects eigenvectors corresponding to lowest eigenvalues is suitable for multiple canaries, as discussed in section 5 of [6].
> >
> > By no mean I claim this method is superior to the one discussed in this work, but I find the lack of comparison to an existing baseline method a downside of the current state of this work.
> >
> > [5] Jagielski, Matthew, Jonathan Ullman, and Alina Oprea. "Auditing differentially private machine learning: How private is private sgd?." Advances in Neural Information Processing Systems 33 (2020)
> >
> > [6] Pillutla, Krishna, et al. "Unleashing the power of randomization in auditing differentially private ml." Advances in Neural Information Processing Systems 36 (2023)

---

> > > ### Author Response · Authors · 2025-11-22
> > > **Clarification and Additional Baseline**
> > >
> > > To clarify, our metagradient optimization method does not require access to the final model being audited. We instead train a surrogate model (ResNet-9 without DP) at each metagradient step to optimize the canaries (thus, the optimization can be completed even before the final DP model is ever trained). We show then that our canaries are effective across a variety of choices for the final audited model: various privacy budgets, model architectures (e.g. WRN 16-4 and now, ViT-Tiny, for CIFAR-10), and training initializations (from scratch or after public pretraining). In this case, the white-box canaries in C.3 of [1] do not apply to the black-box threat model we study in this paper. As an aside, we initially did consider whether these two white-box input canaries can be used as additional baselines, but as [1] states, “it is not trivial to extend either the adversarial example or [their] crafting approach to the black-box setting and therefore [they] do not use them in [their] black-box experiments.”
> > >
> > > We agree that ClipBKD could be used. We originally did not consider this method since [6] assumes that for ClipBKD to be effective, the dimensionality (d = 32x32x3 = 3072) of the data should be much larger than the number of canaries (m = 1000). However, we agree with your feedback and have run ClipBKD for CIFAR-10 at epsilon=8 (see table below).
> > >
> > > In our final revision, we will include ClipBKD as a baseline for all our experiments on CIFAR-10 and CIFAR-100 (we will exclude MNIST and Fashion-MNIST, where d = 28x28x1 = 784 < m = 1000).
> > >
> > > | **Audit Procedure** | **Canary Type**        | **(a) DP Training Avg.** | **(b) DP Finetuning Avg.** |
> > > |---------------------|-------------------------|---------------------------|-----------------------------|
> > > | **(1) Steinke et al. (2023)** | random                 | 0.285                     | 0.477                       |
> > > |                     | random mislabeled       | 0.308                     | 0.489                       |
> > > |                     | ClipBKD      | 0.129                     | 0.487                       |
> > > |                     | metagradient (*ours*)   | **0.392**                 | **0.665**                   |
> > > | **(2) Mahloujifar et al. (2024)** | random                 | 0.405                     | 0.687                       |
> > > |                     | random mislabeled       | 0.225                     | 0.632                       |
> > > |                     | ClipBKD      | 0.270                     | 0.841                       |
> > > |                     | metagradient (*ours*)   | **0.732**                 | **1.207**                   |

---

> > > > ### Comment · Reviewer_Vbyr · 2025-11-22
> > > >
> > > > Isn't the surrogate model idea orthogonal to the canary crafting method?
> > > >
> > > > If I understand correctly, any canary construction method can be carried over other models, hoping the quality of the resulting canaries is "transferable", just like you did here with the ClipBKD technique.
> > > >
> > > > It seems to me like this work contains two separate contributions. The main one is a novel canary construction method in the final model setting (IV.B) using meta-gradients, which should be compared to relevant baselines such as ClipBKD and others (some of whom I mentioned in previous comments). The other one is the empirical observation that (at least some) canaries constructed using one model are effective for auditing other models, thus saving the need to assume final model access.
> > > >
> > > > I will increase my score, but strongly recommend the authors better situate this method among the existing baseline methods, rather than as the first method for canary optimization under this assumptions setting.

---

> ### Author Response · Authors · 2025-12-03
> **Response**
>
> We thank you for updating your score.
>
> We think it’s useful to clarify that we can identify in our work two Separate phases
>
> **Phase 1: Canary Construction (Pre-Auditing)**:
> Before any auditing takes place, we craft canaries by running meta-gradient optimization on a surrogate ResNet-9 model using non-private SGD. We consider this is an offline preprocessing step that produces a fixed set of canary images. , Referring to your comment about “final model access”, we clarify that this phase requires no access to the final model being audited and must occur before training on the audited model begins. This is true for any canary construction method (e.g., mislabeling images, ClipBKD, etc.)  in the black-box setting ,
>
> **Phase 2: Privacy Auditing (Final Model Access Only)**:
> Once the canaries are constructed, we conduct the actual auditing procedure. Here, a random subset of the pre-crafted canaries is inserted into the training set. The auditor then waits for the target model to be trained and then observes only the final trained weights. The auditor performs membership inference using only the model's output predictions on the canaries. At no point during this phase does the auditor access intermediate checkpoints, gradients, or any internal training state.​
>
> In addition, we would like to comment on two other points.
>
> [**Transferability is not a contribution**] Rather than be orthogonal, the surrogate model optimization is simply our method for constructing effective canaries—analogous to how ClipBKD uses SVD to construct canaries robust to DP-SGD. Both ClipBKD and our method produce a fixed canary set before auditing begins, then conduct auditing under identical final-model-access assumptions. The "transferability" observation is not a separate contribution but rather an empirical validation that our construction method works. As you have suggested, one would hope that any canary construction method possesses this "transferability" quality, and our experiments validate this to be true for our method.
>
> [**Situating among prior baselines**] While our work is not the first to study canary optimization for DP auditing in general, we emphasize that it is the first to study it in the **one-run**, black-box/last-iterate setting. Our experiments show that our method outperforms prior baselines studied in this setting. Following your feedback, we have also now shown that our method outperforms methods borrowed from other settings (i.e., ClipBKD from the **multi-run** auditing literature).

---

### Official Review · Reviewer_tQae · 2025-10-28

**Soundness:** 2
**Presentation:** 3
**Contribution:** 2
**Rating:** 4
**Confidence:** 4

**Summary:**

This paper studies black-box auditing of DP-SGD and proposes a canary-generation method based on metagradient optimization to strengthen empirical privacy lower bounds. The approach uses a proxy (non-private) model to optimize canaries and then applies these optimized canaries to audit models trained via DP-SGD following existing one-run procedures (Steinke et al. 2023 and Mahloujifar et al. 2024). They empirically show on CIFAR10 that metagradient optimized canaries result in stronger attacks than random canaries under a fixed privacy budget.

**Strengths:**

- The paper is well-presented and clearly situates itself within the literature of DP (black-box) auditing and canary-based attacks.
- The explicit focus on last-iterate black-box auditing is an important and practically relevant problem and the idea of using optimized canaries to audit in this setting is an open problem.

**Weaknesses:**

- Empirical results are limited as only a single dataset (CIFAR10) is used alongside one non-DP experiment and one DP setting (with epsilon=8). It remains unclear how these gains using optimized canaries generalize to other datasets or differing privacy budgets, particularly high-privacy regimes.
- The paper claims efficiency of the canary optimization approach (using REPLAY) but does not quantify these overheads empirically or otherwise.
- This idea of using optimized canaries seems a straightforward extension of attacks in alternative settings (white-box auditing).

**Questions:**

1. The main contribution of the paper does not seem to be the optimized canary approach since this has been used before in prior work [1,2] but in adapting this to a black-box setting via a separate surrogate model/loss. However, there does not seem to be much explanation or detail as to why and when this proxy model approach would be effective vs. using random canaries.
2. It is not clear how the proxy model is trained and how much of an effect this has on the DP-SGD auditing process. Could you give more information on the precise training setup that is being used in your experiments?
3. One of the contributions is leveraging REPLAY to provide efficient metagradient computations. However, there are no results that highlight the overhead of crafting these optimized canaries. The tradeoffs between this and random canaries is not made clear i.e., why bother with optimized canaries when random canaries could be good enough for the amount of compute needed? How practical is the suggested attack?
4. There are no results on differing privacy budgets or different datasets which limits how strong a conclusion to takeaway on the current set of results. Does your approach work well for auditing other datasets and model architectures?
5. How do the learned canaries look? Are they diverse or do they collapse to similar patterns as random noise? How are canaries initialized?
6. The paper dedicates a substantial portion of space to reviewing prior work and setup (~6 pages), leaving only a little room for the proposed method and the experimental results. It would be beneficial to provide more space for the technical details and additional experiments.

[1] Nasr, Milad, et al. "Tight auditing of differentially private machine learning." 32nd USENIX Security Symposium (USENIX Security 23). 2023.

[2] Maddock, Samuel, Alexandre Sablayrolles, and Pierre Stock. "Canife: Crafting canaries for empirical privacy measurement in federated learning." arXiv preprint arXiv:2210.02912 (2022).

---

> ### Author Response · Authors · 2025-11-19
> **Review Response**
>
> Thank you for your constructive comments! Please let us know we can provide any additional information that may help in improving your final score.
>
> **Weaknesses**
>
> - Weakness 1: We have expanded our experiments (see general comment) and plan to include in our revision results on three additional datasets (CIFAR-10, CIFAR-100, MNIST, Fashion-MNIST, as well as evaluations across different privacy budgets ϵ=8,4,2,1. We find that in general, these extended results are consistent with our experiments on CIFAR-10
>
> - Weakness 2: We will report the computational cost of our method in our final version: each metagradient optimization run requires approximately 96-120 hours on a single NVIDIA A100 GPU. We note that while the surrogate model is trained for many steps for each meta-iteration, the computational cost is still relatively small, given that using a tiny ResNet-9 model as the surrogate still produces very effective canaries. For comparison, training the WideResNet model with DP-SGD requires about 12 hours on the same GPU.
>
> - Weakness 3: We respectfully disagree with this characterization. White-box auditing often involves leveraging or interfering with the training dynamics of the model (gradient injection or modification), which can alter the natural learning process. For example, for the white-box canaries used in Steinke et al. and Mahloujifar et al. (which are derived from Nasr et al.), the auditor does not need to even consider what the canary examples actually are since during training (and calculating scores), the gradient is completely replaced by a one-hot vector. In the black-box setting we consider, the canaries must be actual data examples (i.e., image pixels and labels) that are optimized by a process agnostic to the training dynamics of the model.
>
> **Questions**
>
> 1. Our surrogate objective (Equation 3) is specifically designed to directly optimize the loss gap between included and excluded canaries, capturing the key signal that drives successful membership inference in the black-box auditing process. In contrast, random canaries are not tailored to amplify this signal. In mislabeling canaries, the auditor hopes that loss gap widens since such “incorrect” images should theoretically only be classified correctly if it has been seen during training. Our method instead directly tries to optimize this loss gap, and our results demonstrate that this approach consistently achieves tighter bounds.
>
> 2. We will include a comprehensive description of the training setup for the proxy model for each meta-iteration, including the hyperparameters used to train the ResNet-9 model at each metagrad iteration.
>
> 3. Please see the above response to weakness 2 regarding the computational overhead. Note that once optimized once for each dataset, these canaries can be reused to audit multiple architectures, privacy budgets, or training procedure (i.e., from scratch vs. finetuning). To demonstrate this, we have extended our experiments to four datasets and four privacy budgets.
>
>     Regarding your comment about whether “random canaries could be good enough,” we emphasize our approach substantially provides tighter bounds compared to random canaries to the extent that we believe auditing should always be conducted with our canaries instead of random ones. Moreover, the computational costs (above response to weakness 2) of our metagrad optimization procedure is not significantly more than running DP-SGD itself, meaning that if one were to be auditing DP-SGD, then it is likely that the required amount of computational resources for our method is also available.
>
> 4. Please see the above response to weakness 1. In addition, we have also run additional experiments in which we use the same canaries for auditing vision transformers. However, for the sake of time, we were only able to complete experiments on CIFAR-10 for epsilon=8.0.
>
> 5. The canaries are initialized to random images from the dataset to speed up convergence. While diverse, they end up looking like random shapes and patterns to the human eye. We will include examples of the images in our revision.
>
> 6. Thank you for your suggestion. Given our additional experiments, we will move some part of discussion/explanation of prior work to the appendix so that we can include our new results showing the effectiveness of our method on other datasets and privacy budgets.

---

> > ### Comment · Reviewer_tQae · 2025-11-27
> >
> > I thank the authors for their detailed rebuttal.
> >
> > * For Weakness 1, I am glad to see additional experiments across a wider range of epsilon values, and it is encouraging that these appear consistent with the CIFAR-10 results.
> >
> > * For Weakness 2, could you clarify whether the reported training times correspond to optimizing m = 1000 canaries on CIFAR-10? In line with Reviewer qvdv’s comments, this seems like a substantial computational cost compared to simpler baselines such as the suugested max-norm canary.
> >
> > * I am not quite sure I follow the response to Weakness 3. Isn't it the case that the Nasr et al. work also consider input-space canaries via 1) adversarial examples and 2) their own canary construction?
> >
> > * Regarding Q2, I still find the precise setup for optimizing the canary via a surrogate model (ResNet-9) unclear. I do not see these details in the main paper, and it remains difficult to understand the exact setup used.
> >
> > Because of these remaining concerns, my score will remain unchanged.

---

> ### Author Response · Authors · 2025-12-03
> **Response**
>
> We thank the reviewer for their thoughtful feedback. We address the remaining concerns below.
>
> - To clarify, the reported computational cost of "96–120 hours on a single NVIDIA A100 GPU" refers to the optimization of the full set of m = 1,000 canaries. While we acknowledge that our method introduces non-trivial overhead for a single auditing run, we emphasize that the optimized canaries significantly outperform baseline methods to the extent that we believe merits the additional compute.Moreover, we show that these canaries are transferable across different model architectures and privacy budgets. This reusability renders the amortized cost of our approach negligible in practice, particularly when auditing across multiple privacy configurations, architectures, and random seeds—as demonstrated in our experimental evaluation.
> - The input-space canaries that you are referring to are still in the white-box setting, rather than the black-box setting we study. Thus, they are not applicable. To clarify, Nasr et al., consider three distinct threat models:
>    1) White-box access with gradient canaries: The auditor can modify training by viewing the model parameters and privatized gradient at each step and directly change what gradient is used to update the model.
>    2) White-box access with input-space canaries (**the setting you are referring to**): The auditor again can modify training by viewing the model parameters and privatized gradient at each step. However, instead of being able to directly choose the gradient used to update the model, the auditor can only choose what the canary sample looks like at each step.. We emphasize that this is a white-box setting that differs fundamentally from our setting, where we only observe the final, DP-trained model (last-iterate access).
>    3) Black-box access (**the setting we study**): The auditor can only insert canaries at the beginning of training and view the model parameters after training is completed. Unlike the above two white-box settings, the auditor cannot view intermediate checkpoints or modify training (i.e., changing the gradient directly or using input-space canaries to change the gradients indirectly).
>
> ## Canary Optimization Process Explained
>
> The optimization alternates between two nested loops: inner training (training a surrogate model) and meta-optimization (updating the canary pixels/labels based on the trained model's behavior).
>
> **Inner Training Phase**:
> At each metastep, a fresh ResNet-9 model is trained from scratch using standard SGD on the dataset augmented with half of the canaries (the "IN" set). This produces a trained model that has "seen" only the IN canaries during training, while the OUT canaries remain unseen.​
>
> **Meta-Optimization Phase**:
> Once inner training completes, the algorithm evaluates how well the model distinguishes IN canaries from OUT canaries by computing a loss gap: the average loss on IN samples minus the average loss on OUT samples. Intuitively, a good set of canaries should produce a large gap, low loss on IN (memorized) and high loss on OUT (not generalized to).​
>
> The REPLAY method then computes the metagradient, the gradient of this loss gap with respect to every pixel and label of every canary. This gradient indicates how to perturb each canary's pixels and labels to increase the loss gap. The canaries are then updated with separate learning rates for images and labels.​
>
> **Iterative Refinement**:
> This entire cycle repeats for 500 metasteps. Each metastep uses a fresh random split of canaries into IN/OUT sets and a new random model initialization, which ensures the optimized canaries are robust across different training conditions rather than overfitting to a single configuration.
>
> Below we leave a summary table with the details of the hyperparameters used.
>
> | Component            | Parameter             | Value  |
> | -------------------- | --------------------- | ------ |
> | Inner training (SGD) | Epochs                | 18     |
> | Inner training (SGD) | Learning rate         | 0.2    |
> | Inner training (SGD) | Momentum (b1)         | 0.85   |
> | Inner training (SGD) | Weight decay          | 1e-5   |
> | Inner training (SGD) | Batch size            | 250    |
> | Meta-optimization    | Total metasteps       | 500    |
> | Meta-optimization    | Number of canaries | 1000   |
> | Meta-optimization  | Image learning rate   | 0.5    |
> | Meta-optimization  | Label learning rate   | 0.03   |
> | Meta-optimization | β1 (im/lab)           | 0.25   |
> | Meta-optimization  | Weight decay (im/lab) | 1e-3   |

---

### Official Review · Reviewer_qvdv · 2025-10-28

**Soundness:** 2
**Presentation:** 4
**Contribution:** 3
**Rating:** 6
**Confidence:** 4

**Summary:**

The paper describes a method to discover a strong set of canary examples to use for empirical privacy auditing, via optimizing (with metagradient descent) a function that approximates the distinguishability of canaries *included* or *excluded* from training. Experiments demonstrate that these canaries are significantly strong (produce higher empirical epsilons) than random canaries. Each step of metagradient descent requires a model training run, so the technique would be infeasible on a large architecture, but they show that canaries optimized using a smaller architecture are also strong when ported to a larger architecture.

**Strengths:**

The core idea is original and promising. Although the method is too expensive to use on large architectures, the experiments show that it is sufficient to optimize the canary set with a small architecture and port them over. The writing is very clear and precise.

**Weaknesses:**

I have two main concerns regarding soundness and contribution.

First, the claim that the method can be used on a small architecture and generalized to a larger one is critical for the method to be of real utility, and therefore it needs more evidence. Could you experiment with more model sizes, or generalizing to a completely different architecture like ViT? Maybe at least one other data modality, like audio?

Second, I think a simple baseline that is important to compare to would be maximizing the gradient norm of the canary (on the initial model parameters, or perhaps after some training on real data). This would be computationally much lighter than metagradient optimization through the entire training run, and might produce a set of canaries that are nearly as strong as the ones you find.

**Questions:**

Do you have more evidence that it works to optimize canaries on one model and port them to another?

Could you compare to simply maximizing the norm of the canary gradient, or do you have a convincing argument why this will not work?

Could you add a topline to figure 2, showing the maximum possible empirical epsilons producible by that auditing method with that number of canaries? It would be interesting to see if you are hitting that max.

Note: there's some ambiguity in the literature about what constitutes a "black box" audit. For some, that would imply only allowing the adversary API access, not giving it even the final model checkpoint, or ability to compute a loss. In reality there is a spectrum of auditing scenarios depending on the threat model, so it may be best to retire the "black" vs "white" distinction.

Typo: $i$ vs $t$ in Alg 5

---

> ### Author Response · Authors · 2025-11-19
> **Review Response**
>
> Thank you for your constructive feedback and detailed review. We have included our response below. If there is anything else we can address that may influence your decision to revise your final score, please let us know as well.
>
> **Weaknesses**
>
> - Weakness 1: We have extended our experiments to vision transformers (see general comment). We find that our canaries are just as effective when switching from Wide ResNets to vision transformers and estimate an even higher empirical epsilon for DP-finetuning. In addition, we have also run experiments on MNIST and Fashion-MNIST (see general comment), in which the surrogate model for canary optimization is still ResNet-9, but the audited DP model is a medium-sized CNN. We were unable to complete additional experiments for other modalities. However, we note that most of the existing literature on DP auditing focuses on image classification models, for which we make substantial improvements over previous work.
>
> - Weakness 2: While maximizing the gradient norm could indeed provide an interesting comparison, it is not immediately clear how this approach would extend to our setting in which we require a large number of distinct canaries rather than a single one that has been optimized for this gradient-norm maximization objective. In case we have misunderstood your suggestion, would it be possible for you to provide us with a bit more detail on how this baseline could be implemented for generating a diverse set of (m=1000) canaries?
>
> **Questions**
>
> 1. See above response to weakness 1.
>
> 2. See above response to weakness 2.
>
> 3. We assume you mean what the empirical epsilon would be if the auditor guessed all m=1000 canaries correctly. We can add a topline to Figure 2 to indicate this maximum value. We note however, that we are not close to hitting this value, which is around 6 when using the procedure of Steinke et al. and over 10 (not possible for eps=8.0) when using that of Mahloujifar et at.
>
> 4. In the preliminaries section of our paper, we attempted to formalize our setting carefully to ensure that our framework and the assumptions are clearly specified and consistent with prior work (e.g., Nasr et al., Steinke et al., and Mahloujifar et at.). However, we understand that there is still some inconsistency in how the DP auditing literature uses the terms white-box and black-box.
>
> 5. Thank you for the note. We will correct that in our revision.

---

> > ### Comment · Reviewer_qvdv · 2025-11-19
> > **note on gradient-norm max objective**
> >
> > Thank you for your comments. I'll reply in detail later. To clarify about weakness 2, you might get a diverse set of distinct canaries with high gradient norm just by starting from different initializations, whether random values or perhaps a set of diverse real images. It seems unlikely that all initializations would converge to the same image, given the non-convex objective, although that is an empirical question.

---

> > > ### Author Response · Authors · 2025-12-03
> > > **reply to note on gradient-norm max objective**
> > >
> > > Thank you for this suggestion. It could be interesting to test whether this lightweight approach can yield a diverse set of canaries that is effective for auditing. However, we again would like to emphasize that in our view, the computational cost of metagradient optimization in our experiments is not substantially larger than DP-SGD itself. Thus, we strongly believe that the strong empirical improvements justify (and outweigh) the extra compute needed to optimize the canaries. In addition, given that this gradient-norm max method does not appear anywhere in the literature for DP auditing in either our setting (i.e., one-run auditing) or others (e.g., multi-run auditing), we did not compare to it in our paper and argue that the contributions of our work should not be judged relative to new algorithms entirely. Nevertheless, if the reviewers deem it important, we are very happy to study the proposed algorithm in more detail and report the results in our camera ready version.

---

### Official Review · Reviewer_mkZ7 · 2025-10-31

**Soundness:** 3
**Presentation:** 3
**Contribution:** 3
**Rating:** 6
**Confidence:** 5

**Summary:**

This paper addresses the crucial problem of deriving tighter empirical lower bounds for the privacy parameter (ε) in black-box privacy auditing of differentially private learning algorithms. The authors propose a novel method to optimize canary examples using metagradient descent. This optimization is achieved by iteratively updating the canaries using a surrogate model trained on the CIFAR-10 dataset to minimize a surrogate objective function. This function is defined as the loss gap between canaries included in the training set and those held out. The optimized canaries are then deployed to solve the problem of underestimating privacy loss with standard auditing techniques. Through experiments, the authors demonstrate that their method yields higher and more accurate empirical ε values compared to baseline canaries, particularly in the challenging DP finetuning scenario.

**Strengths:**

S1. This paper proposes a novel and effective approach to improve privacy auditing by actively optimizing canary examples. This significantly advances the SOTA in auditing effectiveness, enabling auditors to uncover more accurate privacy leakage estimates.

S2. The optimized canaries prove effective for auditing models trained both from scratch and via DP-finetuning, which establishes the method's utility in different practical settings of differentially private model deployment.

S3. This paper includes detailed algorithms for both the metagradient optimization and the auditing procedures, which ensures the reproducibility of the proposed method.

**Weaknesses:**

W1. The metagradient optimization process involves training a surrogate model multiple times within each meta-iteration. This paper does not discuss the computational overhead of this optimization phase, which could be a practical barrier for scenario with limited resources.

W2. The empirical evaluation is exclusively confined to image classification tasks using CIFAR-10. The generalizability of this canary optimization approach to other data modalities or different machine learning tasks remain unexplored. And this paper lacks discussion or an ablation study on the sensitivity of the results to different surrogate model architectures for the canaries optimization.

**Questions:**

Overall, this paper is well-structured. I have the following concerns:
1. While the paper compares against random and mislabeled canaries, are there other canary optimization techniques that could serve as stronger baselines for comparison? (e.g., the briefly mentioned Jagielski et al. and Nasr et al. in the related work)

2. This paper states that initial canaries are sampled from CIFAR-10. Could the authors elaborate on the impact of this initial sampling? How would initializing the canaries with entirely random noise affect the optimization process and their final effectiveness?

3. This paper uses a fixed canary set size of m=1000. While effective for the tested scenarios, the scalability of the metagradient optimization to a significantly larger number of canaries remains unexplored. It would be beneficial to understand if there is a practical limit to the number of canaries that can be effectively optimized.

4. This paper mainly presents quantitative results. A deeper analysis or visualization of the optimized canaries would provide mechanistic insights into why they are so effective. For example, showing how their appearance or features differ from regular canaries.

---

> ### Author Response · Authors · 2025-11-19
> **Review Response**
>
> Thank you for your constructive feedback and detailed review. We address each comment and question below. If there is anything else that we can provide that may influence your decision to revise your final score, please let us know as well.
>
> **Weaknesses**
>
> - Weakness 1: We will report the computational cost of our method in our final revision: each metagradient optimization run requires approximately 96-120 hours on a single NVIDIA A100 GPU. We note that while the surrogate model is trained for many steps (18 epochs) for each meta-iteration, the computational cost is still relatively small, given that using a tiny ResNet-9 model as the surrogate still produces very effective canaries. For comparison, training the WideResNet model with DP-SGD requires about 12 hours on the same GPU.
>
> - Weakness 2: While our conclusions are limited to image classification, we note that most of the existing literature on DP auditing studies this setting, for which we make substantial improvements over previous work. Only recently has one-run auditing been extended to textual data [1], and we agree that it would be interesting to study canary optimization in that setting for future work.
>
>     Regarding the suggestion to conduct an ablation study exploring alternative architectures, we agree that this represents a valuable direction for future investigation. However, an in-depth study requires a substantial number of additional experiments that we believe fall outside the intended scope of the current work. Using a much larger model architecture may also lead to problems with respect to computational efficiency. Nonetheless, we argue that it is noteworthy that our method yields canaries applicable to wide range of datasets (CIFAR-10, CIFAR-100, MNIST, Fashion-MNIST) and transferable across many architectures (CNN, WideResNet, ViT), even without us having to extensively optimize the architectural choice of the surrogate model.
>
> [1] Panda, Ashwinee, et al. "Privacy Auditing of Large Language Models." The Thirteenth International Conference on Learning Representations.
>
> **Questions**
>
> 1. To the best of our knowledge, there do not exist other works that have proposed better choices of canaries for one-run auditing, other than the random and mislabeled canaries used in Steinke et al., 2023. While Jagielski et al. and Nasr et al. do also propose canaries for black-box auditing, they consider multi-run auditing in which only a single canary must be inserted into training. It is not obvious to us how to extend their techniques into this setting, in which an entire set of different canaries must be available.
>
> 2. Thank you for the thoughtful question. We did not systematically investigate how different initialization strategies affect the resulting canaries. Our intuition is that the optimization landscape is quite complex, so the meta-optimization procedure likely provides a strong local improvement over whichever starting point is used. This also implies that better initialization, if available, could likely be further enhanced by our meta-optimization.
>
> 3. In our current implementation, GPU memory is the main practical bottleneck when increasing the number of canaries. A straightforward workaround is to run the metagradient optimization multiple times with different random seeds and aggregate the resulting canary sets into a larger one. More broadly, however, we do not see a fundamental scalability barrier in the method itself. In principle, one can shard the computation or parallelize the metagradient updates across multiple GPUs, which would allow substantially larger canary sets to be optimized. Our experiments were conducted under typical limited resource constraints (single A100), but with industrial-scale GPU resources, we expect the approach to scale to significantly larger m without conceptual difficulty.
>
> 4. We will include example images of the canaries in the final revision. We note, however, that to our eyes, the images do not resemble real images, and so to the human eye, they may not provide any insight.

---

### Author Response · Authors · 2025-11-19
**Extended Results I**

We thank all the reviewers for their thoughtful reviews and constructed feedback. To address some comments about our empirical evaluation, we have run experiments on additional datasets (CIFAR-100, MNIST, Fashion-MNIST) and privacy budgets (6.0, 4.0, 2.0, 1.0). Like for CIFAR-10, we optimize all canaries using a ResNet-9 model architecture. For CIFAR-100, we audit the same model architecture as for CIFAR-10 (i.e., WRN 16-4). For MNIST and Fashion-MNIST datasets, we audit a simple CNN model, which for these datasets is already capable enough to achieve decent (90+%) accuracy when trained with DP-SGD at eps=8.0.

We find that in general, our canaries outperform the baselines, especially when using the stronger auditing procedure of Mahloujifar et at. For the procedure of Steinke et al., there are some cases at lower values of epsilons in which all the canaries perform roughly the same. Referring to the non-DP experiments (Figure 2a of the main body), we posit that in cases in which the model has been under trained so that memorization is low (i.e., the case when epsilon is low for DP models or the number of training steps is low for non-DP models), the choice of canaries does not heavily impact the auditing results for the procedure of Steinke et al.

---

### Author Response · Authors · 2025-11-19
**Extended Results II**

### **Auditing results applied to the framework of Steinke et al. (2023)**

| Dataset | Training | Canary Type | ε=8 | ε=6 | ε=4 | ε=2 | ε=1 |
|---------|----------|-------------|-----|-----|-----|-----|-----|
| **CIFAR-10** | DP Training | Random | 0.285 | 0.111 | 0.044 | 0.083 | 0.080 |
| | | Random Mislabeled | 0.308 | 0.150 | 0.088 | 0.017 | 0.018 |
| | | Metagradient (ours) | **0.392** | **0.271** | **0.312** | **0.146** | **0.098** |
| | DP Finetuning | Random | 0.477 | 0.322 | **0.323** | **0.200** | 0.065 |
| | | Random Mislabeled | 0.489 | 0.246 | 0.251 | 0.178 | 0.063 |
| | | Metagradient (ours) | **0.665** | **0.584** | 0.315 | 0.157 | **0.153** |
| **CIFAR-100** | DP Training | Random | 0.148 | 0.134 | **0.170** | **0.131** | 0.075 |
| | | Random Mislabeled | 0.165 | 0.194 | 0.140 | 0.126 | **0.143** |
| | | Metagradient (ours) | **0.422** | **0.410** | 0.141 | 0.069 | 0.074 |
| | DP Finetuning | Random | 0.342 | 0.231 | 0.100 | 0.091 | 0.085 |
| | | Random Mislabeled | 0.175 | 0.200 | 0.147 | 0.150 | 0.165 |
| | | Metagradient (ours) | **0.770** | **0.555** | **0.405** | **0.171** | **0.169** |
| **MNIST** | DP Training | Random | 0.212 | 0.200 | 0.195 | 0.214 | 0.217 |
| | | Random Mislabeled | 0.222 | 0.352 | 0.235 | **0.222** | **0.248** |
| | | Metagradient (ours) | **0.392** | **0.438** | **0.280** | 0.147 | 0.150 |
| | DP Finetuning | Random | 0.254 | 0.239 | 0.249 | 0.230 | 0.214 |
| | | Random Mislabeled | 0.366 | 0.319 | 0.425 | 0.304 | **0.227** |
| | | Metagradient (ours) | **0.736** | **0.601** | **0.650** | **0.364** | 0.142 |
| **Fashion-MNIST** | DP Training | Random | 0.124 | 0.125 | 0.143 | 0.149 | 0.120 |
| | | Random Mislabeled | 0.222 | 0.177 | 0.195 | 0.186 | **0.240** |
| | | Metagradient (ours) | **0.358** | **0.419** | **0.480** | **0.246** | 0.085 |
| | DP Finetuning | Random | 0.268 | 0.257 | 0.266 | 0.225 | **0.218** |
| | | Random Mislabeled | 0.124 | 0.136 | 0.114 | 0.129 | 0.155 |
| | | Metagradient (ours) | **0.817** | **0.996** | **0.672** | **0.390** | 0.172 |



### **Auditing results applied to the framework of Mahloujifar et al. (2024)**
| Dataset | Training | Canary Type | ε=8 | ε=6 | ε=4 | ε=2 | ε=1 |
|---------|----------|-------------|-----|-----|-----|-----|-----|
| **CIFAR-10** | DP Training | Random | 0.405 | 0.213 | 0.077 | 0.000 | **0.030** |
| | | Random Mislabeled | 0.225 | 0.086 | 0.019 | 0.025 | 0.000 |
| | | Metagradient (ours) | **0.732** | **0.498** | **0.229** | **0.049** | 0.000 |
| | DP Finetuning | Random | 0.687 | 0.482 | 0.456 | 0.091 | 0.077 |
| | | Random Mislabeled | 0.632 | 0.618 | 0.378 | 0.307 | 0.016 |
| | | Metagradient (ours) | **1.207** | **0.978** | **0.594** | **0.398** | **0.118** |
| **CIFAR-100** | DP Training | Random | 0.209 | 0.030 | 0.013 | 0.064 | 0.002 |
| | | Random Mislabeled | 0.088 | 0.097 | 0.054 | 0.074 | 0.029 |
| | | Metagradient (ours) | **0.736** | **0.573** | **0.337** | **0.120** | **0.083** |
| | DP Finetuning | Random | 0.354 | 0.274 | 0.065 | 0.115 | 0.038 |
| | | Random Mislabeled | 0.187 | 0.123 | 0.017 | 0.085 | **0.115** |
| | | Metagradient (ours) | **1.286** | **0.935** | **0.594** | **0.370** | 0.113 |
| **MNIST** | DP Training | Random | 0.016 | 0.000 | 0.000 | 0.000 | 0.000 |
| | | Random Mislabeled | 0.070 | 0.204 | 0.142 | 0.039 | 0.008 |
| | | Metagradient (ours) | **0.669** | **0.594** | **0.499** | **0.144** | **0.080** |
| | DP Finetuning | Random | 0.321 | 0.182 | 0.229 | 0.098 | 0.058 |
| | | Random Mislabeled | 0.099 | 0.011 | 0.136 | 0.008 | 0.112 |
| | | Metagradient (ours) | **1.465** | **1.366** | **1.260** | **0.304** | **0.117** |
| **Fashion-MNIST** | DP Training | Random | 0.022 | 0.099 | 0.045 | 0.009 | 0.000 |
| | | Random Mislabeled | 0.141 | 0.085 | 0.094 | 0.061 | **0.136** |
| | | Metagradient (ours) | **0.682** | **0.632** | **0.642** | **0.385** | 0.102 |
| | DP Finetuning | Random | 0.056 | 0.093 | 0.058 | 0.179 | 0.002 |
| | | Random Mislabeled | 0.398 | 0.198 | 0.242 | 0.062 | **0.235** |
| | | Metagradient (ours) | **1.483** | **1.342** | **1.401** | **0.517** | 0.136 |

---

### Author Response · Authors · 2025-11-19
**ViT Extended Results**

To demonstrate that our canaries are transferable to architectures besides CNNs and WideResNets, we also evaluate them on vision transformers. Due to time constraints, we evaluated on ViT-Tiny only for CIFAR-10 at epsilon=8.0. We find that like when using WRN 16-4, our canaries significantly outperform the baselines canaries, with the empirical epsilon being even higher for both auditing methods when conducting DP-finetuning.

**Table: ViT-Tiny on CIFAR-10 at epsilon=8.0**

| **Audit Procedure** | **Canary Type**        | **(a) DP Training Avg.** | **(b) DP Finetuning Avg.** |
|---------------------|-------------------------|---------------------------|-----------------------------|
| **(1) Steinke et al. (2023)** | random                 | 0.088                     | 0.092                       |
|                     | random mislabeled       | 0.255                     | 0.131                       |
|                     | metagradient (*ours*)   | **0.420**                 | **0.943**                   |
| **(2) Mahloujifar et al. (2024)** | random                 | 0.093                     | 0.288                       |
|                     | random mislabeled       | 0.176                     | 0.419                       |
|                     | metagradient (*ours*)   | **0.712**                 | **1.742**                   |

---

### Meta-Review · Area_Chair_mxgm · 2025-12-16

**Summary:**

The common requests from reviewers were to test different architectures and audit procedures. The authors delivered these experimental results during the rebuttal period. The authors also rebutted the reviewers' other issues well. Hence, I recommend accepting the paper.

**Reviewer Concerns:**

I think testing different architectures and audit procedures adequately addressed the reviewers' most pressing concerns. It's impressive that they did all the work during the relatively short rebuttal/discussion period.

**Reviewer Scores:**

Two of the reviewers gave a score of 6. Reviewer Vbyr (initial score: 4) wrote "I will increase my score", which makes another 6 (at least).

Reviewer qvdv requested a simple baseline to compare to, "maximizing the gradient norm of the canary, which can be lightweight compared to the metagradient method suggested by the authors". The authors' reply doesn't appear satisfying (to me). But I have no way to know what this reviewer would think of the author's reply and how the scores would have changed. However, I do not think this is such a pressing issue, but I would recommend the authors include this comparison in the final version.

---

### Decision · Program_Chairs · 2026-01-26

Accept (Poster)